# Characteristics of a new UT-type prefabricated joint for rectangular steel tubes under combined axial-moment loading

Menghan Sun, Luyao He, Zailin Yang [ID]*

Harbin Engineering University, Harbin City, Heilongjiang Province, China

* 194008486@qq.com

## Abstract

This study proposes a UT-type prefabricated joint incorporating adjustable sleeves (±5 mm tolerance) and an energy-dissipating segment to address the practical demands for construction tolerance management and plasticity assurance. Through systematic static and cyclic loading tests, the paper examines the mechanical behavior of UT-type joints under the combined axial force and bending moment. The results demonstrate that axial force significantly influences joint performance. Specifically, axial tension enhances the initial stiffness by 42.6% ($\mu = 0.4$), while axial compression reduces the yield moment by 24.5%. Cyclic loading tests confirm the joint's notable energy dissipation capacity. However, high axial forces induce brittle failure risks; thus, practical engineering applications require attention to buckling in the upper connecting plate of the energy-dissipating segment. Additionally, a simplified trilinear restoring force model is proposed, with an error margin of less than 10%. This model adapts to various loading conditions via axial force-dependent parameter adjustments and demonstrates satisfactory accuracy across validations.

## 1. Introduction

In the development of prefabricated building, beam-column joints have consistently constituted a primary research focus, with scholars globally making important contributions to the advancement of innovative prefabricated beam-column joints [1-18]. For example, Li et al. proposed a beam-column hinged joint incorporating replaceable energy dissipation elements and established critical performance parameters such as strength, stiffness, ductility, and hysteretic curves for this joint configuration [3,4]. Additionally, Tong et al. designed three distinct joint types featuring varied connection geometries and investigated their hysteretic behavior under cyclic axial force. Experimental results revealed that the welded interface between the brace and the gusset plate represents a structural vulnerability susceptible to crack initiation [13]. Another substantial contribution stems from Yang et al., who developed a new steel structural assembly joint wherein damage progression is strategically relocated from

**Data availability statement:** All relevant data are within the manuscript.

**Funding:** The costs of the experiments in this work were supported by the National Natural Science Foundation of China (Grant No. 52279128) and Heilongjiang Provincial Natural Science Foundation (Grant No. YQ2022E013). The funders had no role in study design, data collection and analysis, decision to publish, or preparation of the manuscript.

**Competing interests:** The authors declare no conflict of interest.

the joint core region to peripheral zones. This design adheres to the seismic principle of "strong joint, weak member," with enhanced bearing capacity achieved through increased end-plate thickness [14].

In recent years, rectangular hollow section (RHS) structures have gained widespread application in engineering practice due to their excellent flexural and torsional properties [19–20]. However, research concerning precast beam-column joints specific to RHS structures remains relatively limited [21–24]. While most existing research has concentrated on analyzing the seismic performance and energy dissipation capacity of prefabricated steel beam-column joints, they frequently ignore the installation deviations inherent to the actual construction processes [25–28]. Consequently, certain innovative joint configurations face practical implementation challenges in engineering applications, thereby impeding the broader adoption and large-scale implementation of prefabricated structures.

To solve this challenge, an UT-type prefabricated beam-column joint is innovatively proposed in this paper. Characterized by its adjustable sleeve structure, this joint effectively compensates for construction errors. Furthermore, integrating the design philosophy of a dog-bone joint, a weakened section is strategically incorporated within the beam-column connection area. This configuration ensures the preferential formation of plastic hinges during strong seismic events, thereby not only significantly improving the energy dissipation capacity of the joint but also mitigating the failure risk of high-strength bolts.

Critically, in prefabricated buildings, reinforced truss concrete composite slabs are typically connected to steel beams via shear studs [29–32]. Under seismic loading, the horizontal loads transmitted through floor slabs induce considerable axial force effects at the beam-column joints, which cannot be disregarded. In addition, UT-type joints demonstrate applicability in diverse structural configurations, including portal steel frames, space trusses, and systems incorporating inclined beams. In such scenarios, the axial force at beam ends frequently constitutes a critical influencing factor. Therefore, systematic evaluation of the mechanical properties of these joints under combined axial force and bending moment loading conditions is of fundamental importance for enabling their broader implementation in engineering practice.

To improve computational efficiency and advance engineering applicability, a simplified trilinear restoring force model is established based on cyclic loading tests of UT-type joints under axial force. This model accurately captures critical hysteretic characteristics of the joint, including stiffness degradation and strength attenuation, while exhibiting compatibility with conventional finite element analysis software. Consequently, it provides a robust theoretical basis for both engineering design and seismic performance evaluation.

## 2. Design and construction of new joint and testing setup

### 2.1. UT-type prefabricated joint

The UT-type joints, as shown in Fig 1, comprise beams, columns, adjusting sleeves, and energy dissipation connectors. This innovative prefabricated joint system builds upon the traditional square steel tube beam-column structures. An all-bolt connection

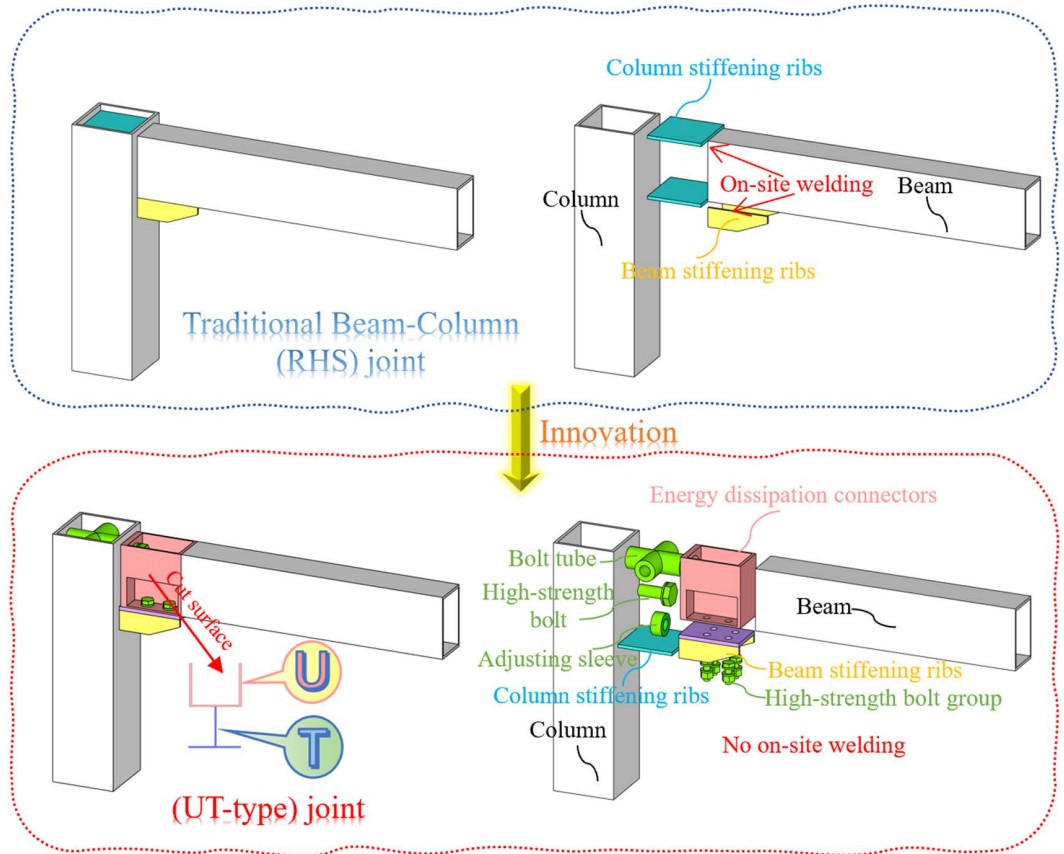

**Fig 1. Structure diagram of the UT-type prefabricated joint.**

is achieved through pre-installation of connecting elements onto the column, while the mitigation of construction errors is facilitated via adjustment of the sleeve-to-column flange interface. In addition, the joint incorporates a strategically positioned energy-dissipating segment between the beam and column (etymologically deriving its "UT" designation from the cross-sectional profile resembling said letters; see Fig 1). By intentionally reducing the bending capacity of this energy dissipation section, plastic hinges preferentially form before high-strength bolts develop excessive stress concentrations. This controlled deformation mechanism substantially improves the energy dissipation capacity of the novel configuration.

Factory-prefabricated units are hoisted to designated elevations (Fig 2(a)). During the prefabrication stage, bolt tubes and beam stiffening ribs are welded onto RHS columns, with corresponding holes precisely drilled at aligned positions to ensure coaxiality with the bolt tubes. Simultaneously, the energy dissipation connectors are welded to the ends of RHS beams. The front end plate of the RHS beam incorporates threaded holes for the adjusting sleeves (Fig 8), ensuring concentric alignment with the axis of the bolt tubes. This manufacturing protocol guarantees precise axial alignment throughout component assembly during on-site installation.

The high-strength bolt group is installed at the lower section of the joint (Fig 2(b)). Importantly, construction deviations induce measurable offset displacement between the energy-dissipating segment and the column.

To compensate, the adjusting sleeve is threaded into the energy dissipation joint area and secured against the column wall (Fig 2(c)). Notably, the sleeve's external threading precisely engages with corresponding threads in the joint region, enabling dimensional adjustment.

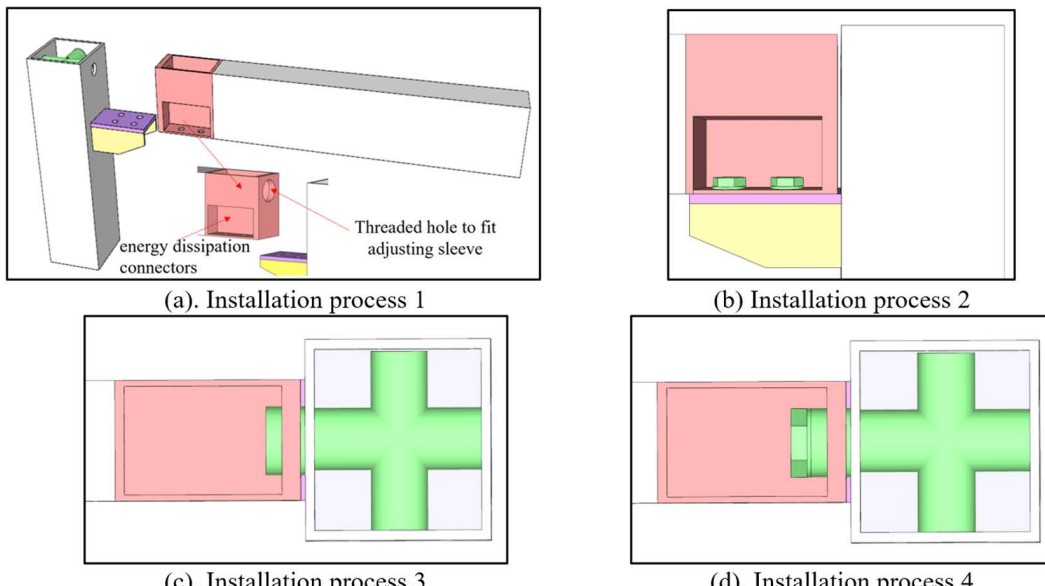

(a). Installation process 1

(b) Installation process 2

(c). Installation process 3

(d). Installation process 4

**Fig 2. Installation process.**

Subsequently, the upper high-strength bolt connection is completed (Fig 2(d)). These bolts require tightening using a calibrated torque wrench, which is inserted into the specialized groove atop the energy dissipation connectors. This configuration facilitates the direct application of the specified pre-tightening force to each bolt. Torque values, predetermined according to design requirements, ensure that the pre-tightening force complies with established structural standards.

## 2.2. Testing setup

Static loading tests were conducted to evaluate the failure modes, bearing capacity, and plastic rotation capacity of UT-type joints. Initially, a specialized beam-column loading test apparatus was engineered in strict accordance with the Code for Seismic Test Methods of Buildings [33]. Through strategic modifications and component integrations, this system enables multi-condition loading simulations across diverse experimental specimens. The physical configuration of the test setup is illustrated in Fig 3.

It is noted that the testing setup employs a beam base rather than connection columns (see Fig 3). This design approach derives from the "strong column, weak beam" principle, wherein failure in UT-type prefabricated joints concentrates at the energy-dissipating connection ends, while the columns exhibit negligible damage. Therefore, the test focuses primarily on the failure mechanisms at these connection ends and within the high-strength bolt groups. Simplification to a beam base configuration preserves test validity while concurrently reducing experimental expenditures.

During static loading tests, specimens undergo preliminary cyclic conditioning comprising two repeated load applications. The imposed load per cycle remains below 10% of the plastic limit moment of the joint, with strict adherence to quasi-static loading protocols. Due to elevated initial stiffness during initial loading phases, force control is implemented incrementally. Prior to achieving the ultimate moment, loading progresses through 50 discrete stages, each applying 0.5 kN·m moment increments with approximately 3-minute stabilization intervals. Upon reaching the ultimate moment, displacement control commences across 10 subsequent stages, each introducing 4 mm displacement increases with approximately 5-minute stabilization periods. Data collection occurs following deformation stabilization at each loading increment.

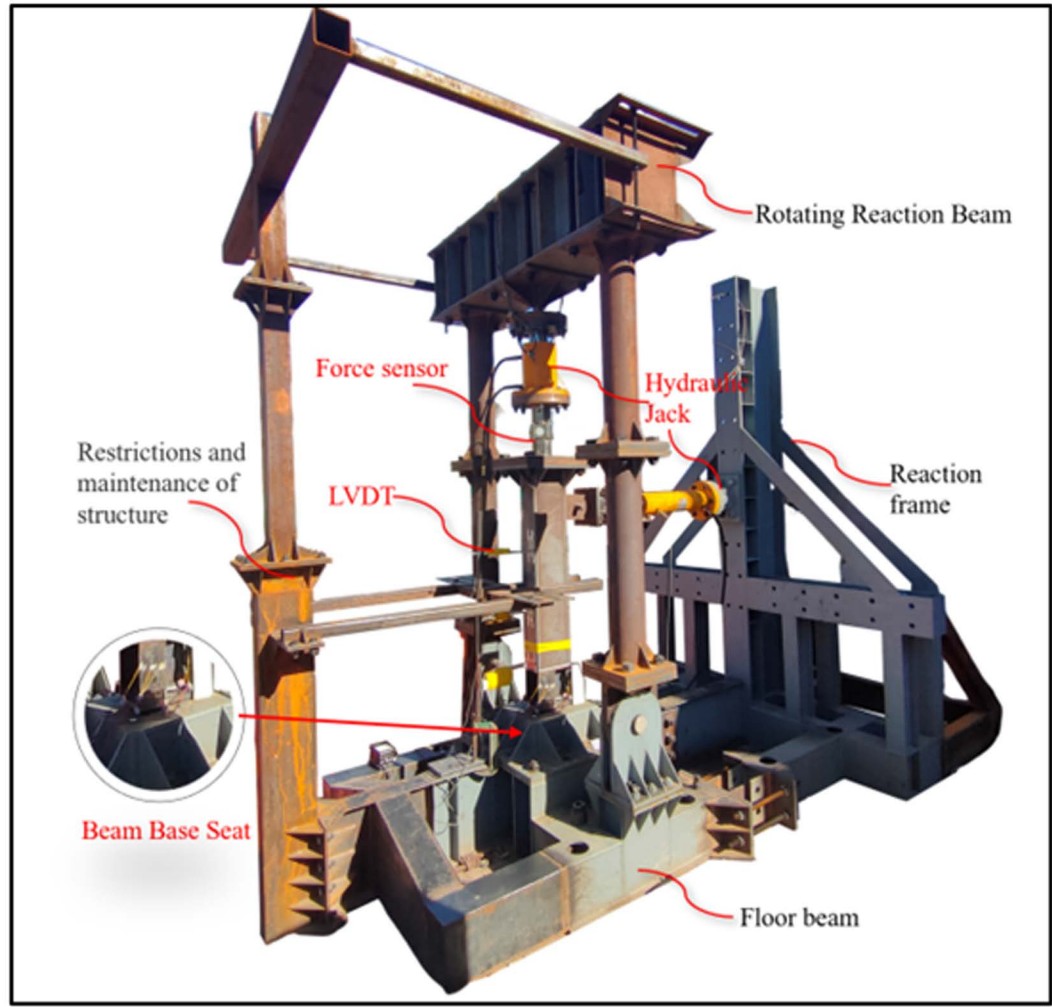

**Fig 3. Field configuration of the structural testing setup.**

In order to systematically examine the influence of axial force on the static behavior of UT-type prefabricated joints, 14 groups of static tests were designed. Table 1 details the corresponding working conditions and the joint identifiers, while Fig 4 illustrates the specimen geometries. For analytical simplification, an axial force ratio ($\mu$) is defined as the absolute axial force magnitude divided by the product of the gross cross-sectional area ($A_g$) and the design value of steel tensile strength ($f_y$), as expressed in Equation 1:

$$\mu = \frac{|N|}{A_g f_y}$$

(1)

## 2.3. Material properties test

The energy-dissipating connections in the designed test specimens utilize Q235 steel with nominal thicknesses of 2.7 mm and 3.7 mm. Uniaxial tensile tests were conducted on both material grades in compliance with standard protocols [34].

**Table 1. Joint number and corresponding working conditions.**

| Joint number | $i$ ($\mu$) | Loading condition | Loading direction |
|---|---|---|---|
| UT-1 | 0 | No axial force | Obverse |
| UT-2 | 0 | No axial force | Reverse |
| UT-(i)ZL-1 | 0.2-0.4-0.6 | Axial tension | Obverse |
| UT-(i)ZL-2 | 0.2-0.4-0.6 | Axial tension | Reverse |
| UT-(i)ZY-1 | 0.2-0.4-0.6 | Axial compression | Obverse |
| UT-(i)ZY-2 | 0.2-0.4-0.6 | Axial compression | Reverse |

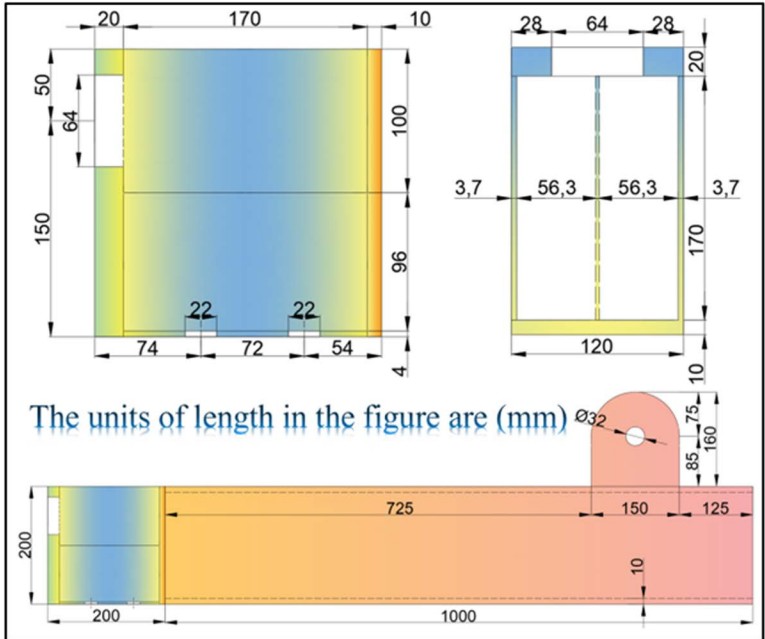

**Fig 4. Geometric dimensions of UT-type joint test specimens (Unit: mm).**

Subsequent mechanical characterization reveals the mechanical properties detailed in Table 2, where the upper yield strength ($R_{eH}$) corresponds to the initial peak stress (i.e., first maximum stress) during yielding, while the lower yield strength ($R_{eL}$) denotes the minimum stress value recorded within the same yield plateau. The tensile strength ($R_m$) represents the material's ultimate capacity, equivalent to the maximum stress attained during the tensile process.

To enhance computational efficiency in simulations, the experimental material property data were simplified using the widely adopted trilinear steel model [35]. This approach defines both the lower yield point ($R_{eL}$) and the ultimate tensile stress ($R_m$) as key inflection points within the stress-strain curve. For Q235 steel at 3.7 mm thickness, $R_{eL}$ was established at 263.5 MPa (corresponding strain = 0.0013), while $R_m$ was determined to be 421.09 MPa (corresponding strain = 0.197).

**Table 2. Mechanical properties of steel materials.**

| Type of steel | $E$ (GPa) | $Rm$ (MPa) | $ReH$ (MPa) | $ReL$ (MPa) | $\delta$% |
|---|---|---|---|---|---|
| 3.7 mm Q235 | 202 | 421.09 | 268.25 | 263.5 | 21.6% |
| 2.7 mm Q235 | 220 | 433 | 315.4 | 308.2 | 58.5% |

Similarly, for 2.7 mm thick Q235 steel, $R_{eL}$ was found to be 308.2 MPa (corresponding strain = 0.0014) and $R_m$ was measured as 433 MPa (corresponding strain = 0.1948).

## 3. Test results

### 3.1 Experimental phenomena

In this experimental investigation, 14 groups of static loading tests were completed. Due to the similar failure modes under different axial compression ratios, and considering the limitations of space. Only the failure characteristics of UT-type joints under axial-force-free conditions and with an axial force ratio (0.4 $\mu$) are presented herein. The specific failure manifestations are detailed as follows:

The failure mode of the UT-type prefabricated joint under positive bending moment is illustrated in Fig 8. Significant deflection occurred in the joint zone, with the rotation angle $\theta$ reaching 0.04 rad. Due to the restraint provided by high-strength bolts, the energy-dissipating connection in Region B underwent local buckling under compression. Conversely, Region A (subjected to tension) exhibited pronounced tensile yielding failure in the energy-dissipating segment. This experimental failure mode aligns with finite element simulation predictions.

Under negative bending moment, as illustrated in Fig 9, the upper side plate of the energy dissipation zone experienced inward buckling failure (see Region A). Region B manifested tensile yielding failure, while the high-strength bolt group at the base exhibited no slippage. At this stage, the joint rotation $\theta$ attained 0.05 rad. Notably, finite element simulations predicted outward buckling in the energy dissipation zone. Nevertheless, this discrepancy can be mainly ascribed to manufacturing deviations in the geometry of the upper side plate of the energy dissipation component.

Fig 10 depicts the failure mode under the coupled 0.4 $\mu$ axial tension and positive bending moment. Distinct from pure bending behavior, the coupled loading suppresses significant warping deformation at the base of the energy-dissipating segment at equivalent rotation angles. The axial tension increases the initial stiffness of the joint by 42.63% compared to pure bending, while its yield moment decreased notably by approximately 6.35%. After node failure, measurements indicated that the gap between the front endplate and the column flange remained approximately 5 mm, with no failure observed in the top high-strength bolt group and no slippage in the lower bolt group of the energy dissipation zone.

The failure mode of the joint subjected to the coupled 0.4 $\mu$ axial tension and negative bending moment is shown in Fig 11. Relative to pure bending loading, a distinguishing feature of this coupled failure mechanism is the induction of significant warping deformation (approximately 5 mm) at the base of the energy-dissipating segment (see Region B) under identical rotation angles. Concurrently, minor slippage was manifested in the lower high-strength bolt group of the energy dissipation zone. The axial tensile force constrains the buckling deformation in the top connecting plate of the energy-dissipating segment, thereby increasing the initial stiffness of the joint by 21.18% compared to pure bending conditions, while reducing the ultimate bending moment by approximately 1.21%. Crucially, the distribution depicted in Fig 25 confirms that the stress within the upper connecting plate remained below the material's yield stress at joint failure.

The failure mode of the UT-type prefabricated joint under coupled 0.4 $\mu$ axial compression and positive bending moment is illustrated in Fig 12. Compared to pure bending loading, the joint deformation at yield failure is markedly reduced. Under this coupled loading, the joint enters the yield stage at a rotation angle of $\theta$ = 0.006 rad—significantly lower than the yield rotation of 0.012 rad under pure bending conditions. Post-failure analysis reveals minimal warping deformation at the front plate periphery and no failure in the high-strength bolts. Mechanistically, the axial compression constrains lateral deformation, thereby increasing the initial stiffness of the joint by approximately 11.7% while reducing its yield moment by about 24.5% relative to pure bending.

The failure mode under coupled 0.4 $\mu$ axial compression and negative bending moment is depicted in Fig 13. At a joint rotation of $\theta$ = 0.001 rad, the upper connecting plate of the energy-dissipating segment undergoes instability-induced buckling, resulting in an abrupt approximately 16.16% reduction in structural stiffness. Concurrently, the yield bending moment decreases by about 9.2% compared to pure bending. As evidenced by the failure morphology (Region A in Fig 8(d)), the

upper region of the energy-dissipating segment experienced buckling failure, whereas the lower region exhibited no significant failure characteristics.

### 3.2. Data analysis

In accordance with the LRDF specification, the yield moment ($M_y$) of the joint under combined axial force and bending moment loading is determined. The computational results are presented in Table 3. The data indicate that the $M_y$ values obtained from the LRDF-based calculations demonstrate conservatism under axial tension conditions. Conversely, under axial compression, manufacturing imperfections within the energy-dissipating segment induce premature buckling failure in the upper connecting plate. Therefore, theoretical $M_y$ values exceed those measured experimentally. The average deviation between the experimental results ($M_y$, test) and theoretical predictions is about 10%, thereby validating the engineering applicability of the LRDF code for joint strength assessment.

Analysis of Table 3 further reveals that the axial force contributes to a measurable enhancement of the joint's initial stiffness ($K_y$). Notably, under axial tension, the $K_y$ value for specimen group UT-0.4ZL-1 exhibits an approximate 42.6% increase compared to UT-1; under negative bending moment loading, the corresponding increase is about 21.1%. In contrast, the variation in $K_y$ under axial compression is comparatively modest, differing by 11.7% and 16.1% from pure bending loading conditions. However, as illustrated in Figs 5–7, the $M_y$ value decreases markedly with escalating axial force magnitude. This inverse relationship between increased $K_y$ and diminished $M_y$ necessitates consideration of potential joint "brittle failure" when subjected to significant axial forces.

## 4. Establishment of numerical simulation model and comparison with test results

### 4.1. Numerical simulation model

To further investigate the mechanical performance of UT-type prefabricated joints and conduct parameter analysis, it is imperative to establish corresponding numerical simulation models. This paper employs ABAQUS software for joint simulation, with model dimensions strictly consistent with experimental specimens (refer to Figs 8–9). The joint components

**Table 3. Characteristic values of UT-type joint force performance.**

| Joint ID | $Ky$ (kN·m/rad) Experimental | $My$ (kN·m) Experimental | $My$ (kN·m) Theoretical value | Rate (%) |
|---|---|---|---|---|
| UT-1 | 3440.16 | 25.1 | 27.07 (308 Mpa) | 7.3%↑* |
| UT-2 | 3879.31 | 20.0 | 21.7 (263 Mpa) | 8.2%↑* |
| UT-0.2ZL-1 | 3204.28 | 22.43 | 23.2 (308 Mpa) | 3.4%↑* |
| UT-0.2ZL-2 | 3652.58 | 18.32 | 19.8 (263 Mpa) | 8.0%↑* |
| UT-0.2ZY-1 | 2783.80 | 18.56 | 21.6 (308 Mpa) | 16.3%↑* |
| UT-0.2ZY-2 | 2867.69 | 16.69 | 18.4 (263 Mpa) | 10.2%↑* |
| UT-0.4ZL-1 | 6333.33 | 23.6 | 22.8 (308 Mpa) | 3.5%↓* |
| UT-0.4ZL-2 | 4700.85 | 19.76 | 19.5 (263 Mpa) | 1.3%↓* |
| UT-0.4ZY-1 | 4,959.88 | 18.94 | 22.4 (308 Mpa) | 15.4%↑* |
| UT-0.4ZY-2 | 3,252.48 | 18.16 | 19.2 (263 Mpa) | 5.4%↑* |
| UT-0.6ZL-1 | 5195.83 | 12.47 | 10.8 (308 Mpa) | 13.3%↓* |
| UT-0.6ZL-2 | 4909.09 | 10.5 | 9.2 (263 Mpa) | 12.3%↓* |
| UT-0.6ZY-1 | 1704.87 | 8.92 | 8.9 (308 Mpa) | 0.2%↓* |
| UT-0.6ZY-2 | 1750.00 | 7.35 | 7.6 (263 Mpa) | 3.4%↑* |

(↑*) represents the ratio of the increase in theoretical data compared to experimental data.

(↓*) represents the ratio of the decrease in theoretical data compared to experimental data.

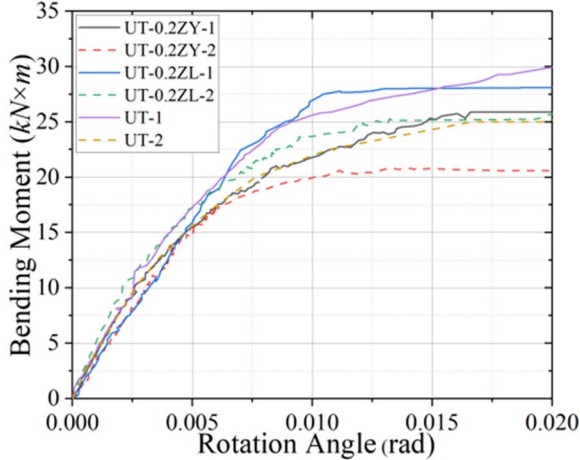

**Fig 5. Loading moment-angle curve in static testing (0.2 μ).**

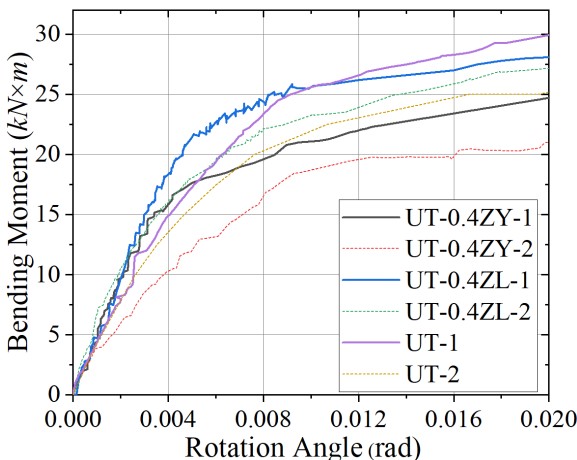

**Fig 6. Loading moment-angle curve in static testing (0.4 μ).**

were discretized using solid elements (C3D8R), and structured meshes were uniformly applied throughout the joint model. To mitigate stiffness distortion induced by single-layer mesh integration during computation—which may precipitate premature local buckling failure—at least three layers of structural meshes were allocated along the thickness direction during meshing. Furthermore, localized mesh refinement was implemented at critical sections, such as energy-dissipating connections, to preserve stress information while enhancing computational efficiency, as illustrated in Fig 14.

During loading simulation, column ends were modeled as hinged supports. Contact interfaces between the gasket and bolt, as well as between the gasket and base plate, were configured to replicate actual engineering conditions. The Penalty friction contact method was universally adopted with a friction coefficient of 0.3. Due to the prohibitive computational cost associated with explicit thread engagement simulation, the *Tie* constraint was applied to model threaded connections; this approach was extended to the outer surface of the adjustment sleeve. Notably, since the inner diameter of the adjustment sleeve exceeds the screw diameter, contact behavior between them remains absent until substantial screw deformation occurs. Consequently, the *General Contact* algorithm was employed to capture this mechanical response.

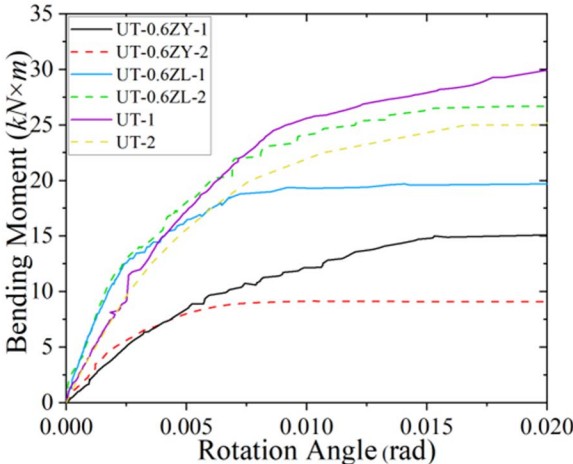

**Fig 7. Loading moment-angle curve in static testing (0.6 μ).**

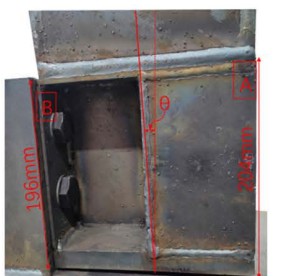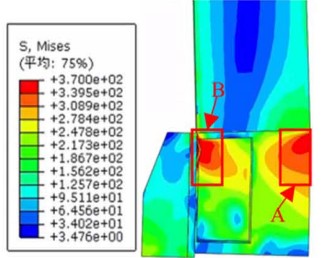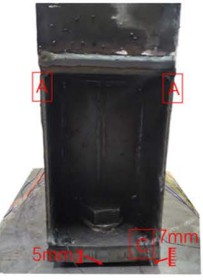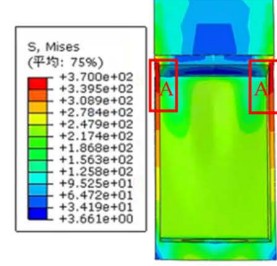

**Fig 8. Positive bending moment loading failure diagram (No axial force).**

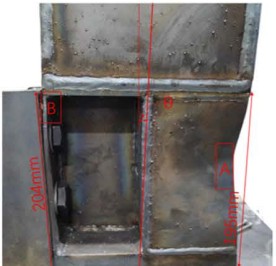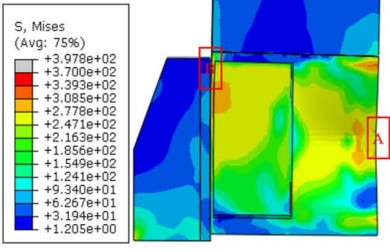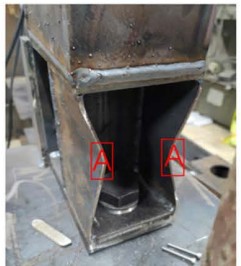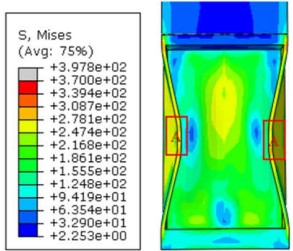

**Fig 9. Negative bending moment loading failure diagram (No axial force).**

The numerical model was discretized along the plate thickness direction, with location-specific material parameters assigned to each segment. Material properties of the energy-dissipating connection segments correspond to the simplified trilinear model derived from experimental data. For steel plates lacking thickness measurements in tests, material parameters comply with Table 4.4.1 of the "Code for Design of Steel Structures" [36]. This code specifies a bilinear stress-strain relationship for Q235 steel, with yield stress ($f_y$) of 235 MPa, ultimate stress ($f_u$) of 370 MPa, elastic modulus ($E$) of 200 GPa, and post-yield hardening modulus set to 0.01 times the initial elastic modulus [37–39].

Geometric parameters and material properties of high-strength bolts were defined according to "High-Strength Large Hexagon Nuts for Steel Structures" and "High-Strength Bolts for Steel Structures" standards [40–41]. Per these specifications, the bolt material exhibits a yield strength $f_y = 940$ N/mm² and ultimate tensile strength $f_u = 1040$ N/mm². Preload forces were applied to match actual engineering demands: 327.6 kN for M30 bolts and 142.7 kN for M20 bolts [42]. The loading sequence followed the experimental protocol: axial force was applied initially, and bending moment increments were introduced in subsequent analysis steps.

## 4.2. Comparison with test results

In order to verify the reliability of the finite element model, identical loading conditions to those in the static tests were applied in simulations. The comparative results are shown in Table 4. The analysis demonstrates that the simulated values of $K_y$ and $M_y$ exhibit strong agreement with experimental measurements. Additionally, it is noted that the relative error of $K_y$ increases proportionally with the axial force ratio. This phenomenon may be attributed to the widening contact gap between the external threads of the adjustment sleeve and the threads of the front end plate under axial compression. This geometric discontinuity reduces synergistic load transfer between components, consequently causing significant deviations in $K_y$ between simulation and experimental results. Fig 7 shows that the finite element simulations accurately replicate the spatial distribution of damage locations observed during physical testing, thus validating the model's accuracy from the perspective of failure mode progression.

## 5. Cyclic loading of finite element model

### 5.1. Design of cyclic loading test

Three specimens with distinct geometric dimensions were designed to systematically evaluate the energy dissipation performance of UT-type joints under axial loading (Fig 15). Utilizing finite element analysis, multiple cyclic loading protocols were configured for each specimen to conduct comparative studies. The loading conditions primarily include: (1) a zero-axial-force state; (2) three characteristic axial force ratios (0.1 $\mu$, 0.2 $\mu$, and 0.3 $\mu$). All specimen identifiers and their corresponding loading parameters are summarized in Table 5. The numerical modeling method is consistent with the static test modeling procedures described in preceding sections.

Table 4. Characteristic values of UT-type joint force performance by simulation.

| Joint ID | $K_y$ (kN·m/rad) Simulated | Rate(%) | $M_y$ (kN·m) Simulated | Rate (%) |
|---|---|---|---|---|
| UT-1 | 4019.29 | 9.4%↓ | 27.5 | 8.7%↑ |
| UT-2 | 3834.22 | 1.2%↓ | 21.2 | 5.6%↑ |
| UT-0.2ZL-1 | 3190.06 | 0.4%↑ | 22.82 | 1.7%↓ |
| UT-0.2ZL-2 | 3719.21 | 1.8%↑ | 18.10 | 1.2%↓ |
| UT-0.2ZY-1 | 2901.56 | 4.2%↓ | 17.13 | 7.7%↓ |
| UT-0.2ZY-2 | 2914.89 | 1.6%↓ | 16.44 | 1.4%↓ |
| UT-0.4ZL-1 | 3882.59 | 8.7%↑ | 21.8 | 8.2%↓ |
| UT-0.4ZL-2 | 5090.91 | 8.3%↑ | 18.16 | 8.8%↓ |
| UT-0.4ZY-1 | 4,57.48 | 16.2%↓ | 17.56 | 7.8%↓ |
| UT-0.4ZY-2 | 2662.72 | 18.1%↓ | 17.50 | 3.7%↓ |
| UT-0.6ZL-1 | 4802.20 | 7.5%↑ | 11.75 | 5.7%↓ |
| UT-0.6ZL-2 | 4126.21 | 15.9%↑ | 11.5 | 9.5%↓ |
| UT-0.6ZY-1 | 1981.56 | 16.2%↓ | 7.25 | 18.7%↓ |
| UT-0.6ZY-2 | 2098.21 | 19.8%↓ | 7.05 | 4.0%↓ |

(↑) represents the ratio of the increase in simulated data compared to experimental data.
(↓) represents the ratio of the decrease in simulated data compared to experimental data.

**Table 5. UT-type joint numbers for cyclic loading.**

| Joint number | $i$ | $j$ | Axial force ratio/k | Loading condition/ (Axial force) |
|---|---|---|---|---|
| $U_4$-$D_4$ | 4 | 4 | 0 | None |
| $U_4$-$D_4$-T-$k_\mu$ | 4 | 4 | 0.1;0.2;0.3 | Tension |
| $U_4$-$D_4$-C-$k_\mu$ | 4 | 4 | 0.1;0.2;0.3 | Compression |
| $U_4$-$D_5$ | 4 | 5 | 0 | None |
| $U_4$-$D_5$-T-$k_\mu$ | 4 | 5 | 0.1;0.2;0.3 | Tension |
| $U_4$-$D_5$-C-$k_\mu$ | 4 | 5 | 0.1;0.2;0.3 | Compression |
| $U_5$-$D_5$ | 5 | 5 | 0 | None |
| $U_5$-$D_5$-T-$k_\mu$ | 5 | 5 | 0.1;0.2;0.3 | Tension |
| $U_5$-$D_5$-C-$k_\mu$ | 5 | 5 | 0.1;0.2;0.3 | Compression |

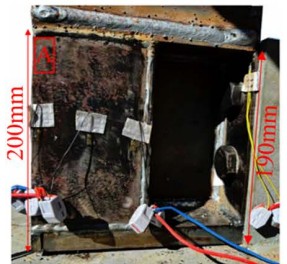 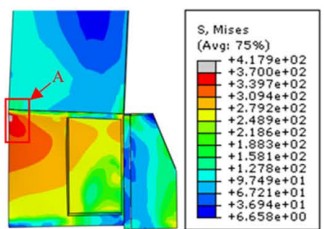 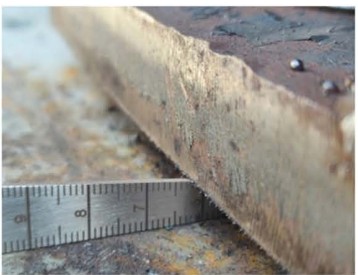 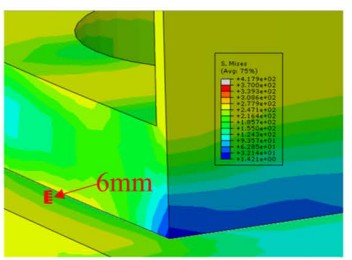

**Fig 10. Failure mechanism diagram under coupled axial tension and positive moment ($0.4\ \mu$).**

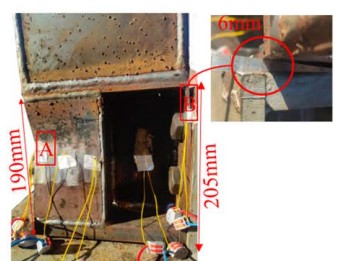 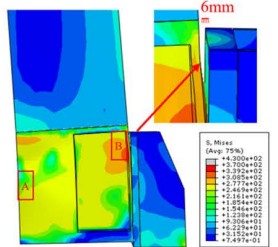 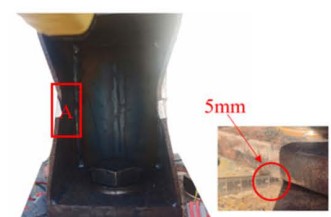 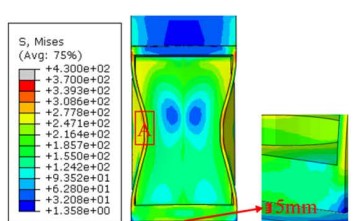

**Fig 11. Failure mechanism diagram under coupled axial tension and negative moment ($0.4\ \mu$).**

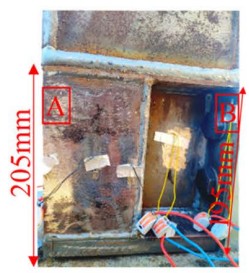 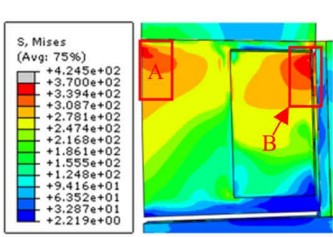 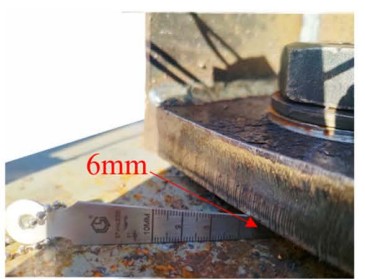 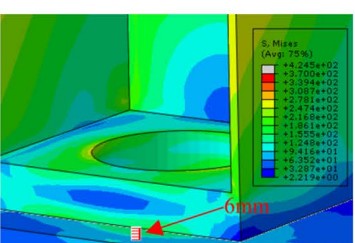

**Fig 12. Failure mechanism diagram under coupled axial compression and positive moment ($0.4\ \mu$).**

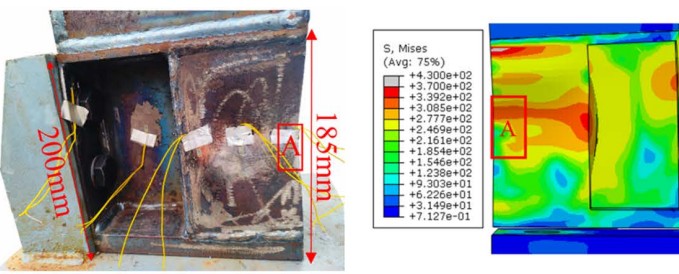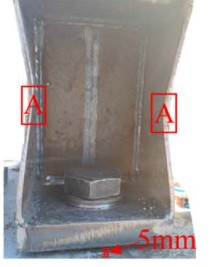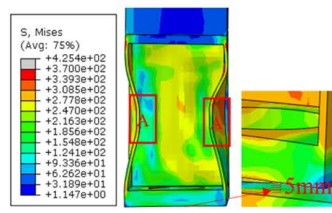

**Fig 13. Failure mechanism diagram under coupled axial compression and negative moment ($0.4\ \mu$).**

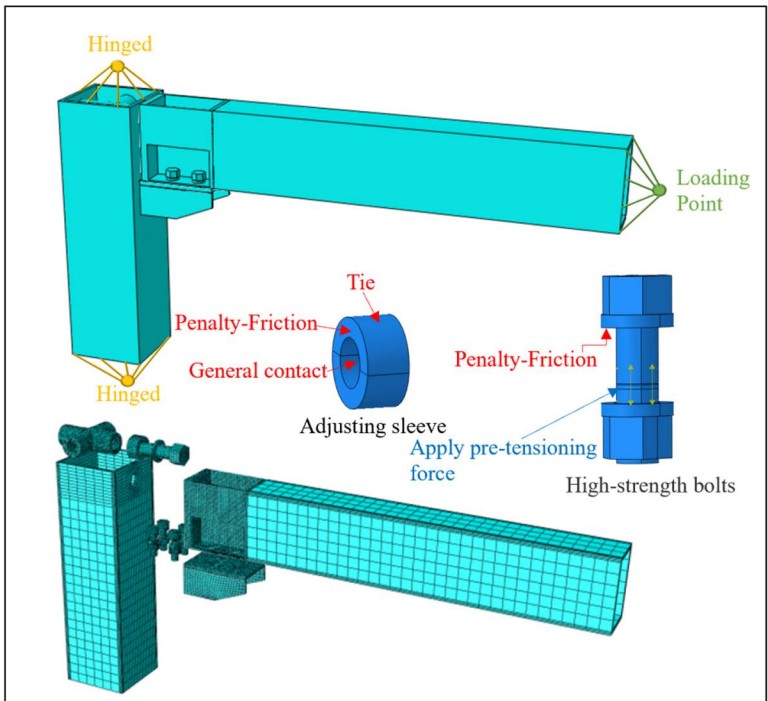

**Fig 14. UT-type joint finite element model.**

According to the relevant specifications [30], cyclic loading tests were conducted on UT-type joints using a force-displacement hybrid control protocol. The experimental procedure comprised three sequential phases: (1) Pre-yield force-controlled loading with three identical cycles per load increment; (2) Post-yield displacement-controlled loading, where displacement amplitudes were scaled by integer multiples ($1\Delta_y$, $2\Delta_y$, $3\Delta_y$...) based on the yield displacement $\Delta_y$ determined from static tests, with each amplitude level cycled three times; and (3) Continuation of loading until either the bearing capacity of the specimen degraded to 85% of the peak load or catastrophic failure occurred (see Fig 16 for the loading schematic). This systematic approach captures the complete hysteretic response evolution—from elastic behavior through yielding to ultimate failure—thereby delivering critical datasets for quantitative analysis of joint energy dissipation capacity and damage progression mechanisms.

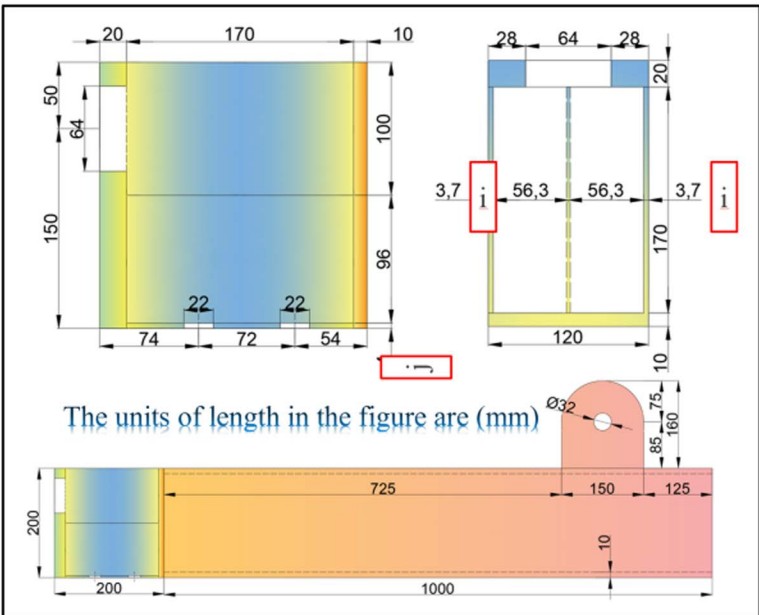

**Fig 15. UT-type joint specimen under cyclic loading.**

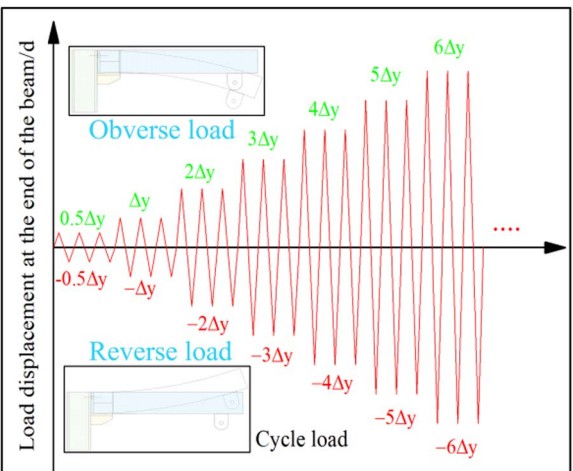

**Fig 16. Cyclic loading system.**

## 5.2. Hysteretic Behavior of UT-type Joints without Axial Force

Fig 17-19 presents the hysteretic curves and corresponding skeleton curves of UT-type joints with varying geometric dimensions under zero axial force conditions. The key mechanical characteristic points extracted from these curves are comprehensively tabulated in Table 6, which includes the yield moment and yield rotation for each joint configuration, both determined via the energy equivalence method.

The hysteretic curves of UT-type joints with varying geometric dimensions reveal two distinct symmetry patterns. The first pattern, exhibited by $U_5$-$D_5$ and $U_4$-$D_4$ joints, demonstrates minimal discrepancy between positive and negative

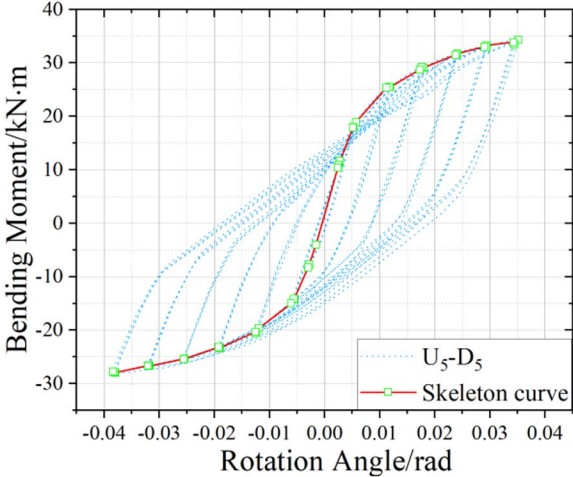

**Fig 17. U$_5$-D$_5$ hysteresis curve.**

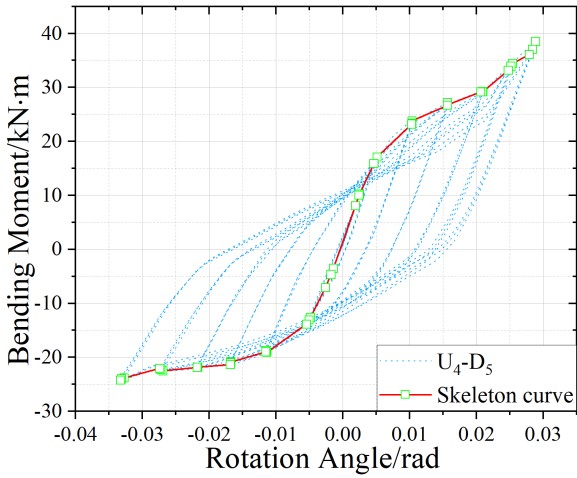

**Fig 18. U$_4$-D$_5$ hysteresis curve.**

moment capacities (bending moments), resulting in symmetric hysteresis loops. In contrast, the U$_4$-D$_5$ joint exhibits significant upward-rightward asymmetry due to the reduced thickness of its upper connecting plate in the energy-dissipating connection, which substantially diminishes its negative moment capacity. Quantitative analysis of characteristic points confirms this observation: the absolute difference in yield moment ($M_y$) for the U$_4$-D$_5$ joint reaches 30.1%, whereas the corresponding differences for U$_4$-D$_4$ and U$_5$-D$_5$ joints are 15.7% and 18.5%, respectively. To enhance structural reliability, engineering design should prioritize minimizing bidirectional moment capacity discrepancies in such connections.

A secondary categorization emerges from the buckling response of the upper connecting side plates. The U$_5$-D$_5$ joint, which experiences negligible plate buckling, maintains stable stiffness during reverse loading cycles and exhibits superior cumulative energy dissipation. Conversely, the U$_4$-D$_4$ and U$_4$-D$_5$ joints undergo progressive stiffness degradation during reverse loading (evident in the lower-left quadrant of their hysteresis curves), especially beyond the fourth loading stage. This degradation manifests as a tendency toward horizontal linearity in the hysteresis loops, ultimately constraining the further development of the energy dissipation capacity of the joints.

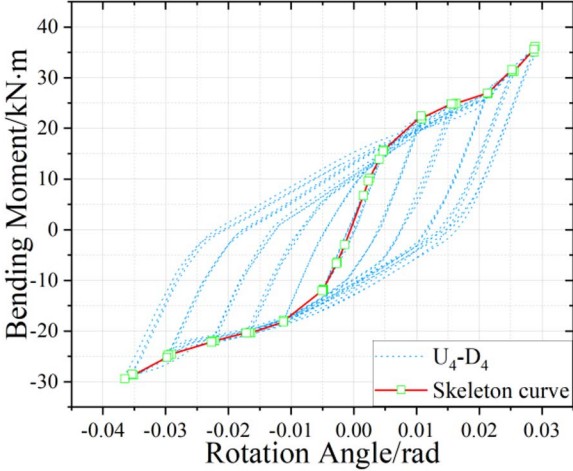

**Fig 19. U$_4$-D$_4$ hysteresis curve.**

**Table 6. Characteristic points of hysteretic curves of each joint without axial force.**

| Joint number | My (kN·m) | θy (rad) | Mmax (kN·m) | θmax (Rad) | Ductility |
|---|---|---|---|---|---|
| U$_5$-D$_5$ | 28.47 | 0.0168 | 34.36 | 0.0351 | 2.089 |
| | −23.20 | −0.0192 | −27.96 | −0.0384 | 2.001 |
| U$_4$-D$_5$ | 29.50 | 0.0212 | 38.52 | 0.0289 | 1.359 |
| | −20.61 | −0.016 | −24.22 | −0.0333 | 2.082 |
| U$_4$-D$_4$ | 27.32 | 0.0217 | 36.22 | 0.0289 | 1.333 |
| | −23.01 | −0.0249 | −29.38 | −0.0365 | 1.462 |

The average ductility factor of the U$_5$-D$_5$ joint reached the highest value of 2.05, whereas those of the U$_4$-D$_5$ and U$_4$-D$_4$ joints are relatively lower at 1.72 and 1.40, respectively. In order to ensure adequate energy dissipation capacity, the upper side plate thickness within the energy-dissipating connections must satisfy the minimum anti-buckling threshold prescribed by design codes. For analytical purposes, such plates should be modeled as fixed-end axially compressed members to accurately capture their stability limits.

To investigate the plastic development across incremental loading levels, the secant stiffness of each joint configuration was computed at discrete load stages using Equation (2):

$$K_i = \frac{|M_i^+| + |M_i^-|}{|\theta_i^+| + |\theta_i^-|}$$

(2)

where $K_i$ denotes the secant stiffness at the $i$-th load cycle; $M_i^+$ and $M_i^-$ represent the peak bending moments during the $i$-th positive and negative loading phases, respectively; while $\theta_i^+$ and $\theta_i^-$ correspond to the joint rotations at the aforementioned peak moment points for positive and negative loading directions, respectively.

According to the data presented in Table 7, the stiffness degradation of each joint exhibits its most pronounced severity during the three-level displacement loading phase ($2_{\Delta y}$), characterized by an average reduction of approximately 45%. This significant degradation unequivocally indicates the onset of irreversible plastic damage at this loading stage. Therefore, to satisfy the design principle of "No damage under frequent earthquakes" in practical engineering applications, the

**Table 7. The average stiffness of each joint under each level of loading without axial force.**

| Joint number | Load level | Average stiffness (kN·m/rad) | Degradation rate (%) |
|---|---|---|---|
| U$_5$-D$_5$ | 1 | 3526.69 | 0 |
| | 2 | 2953.24 | 16.3% |
| | 3 | 1908.33 | 45.9% |
| | 4 | 1424.38 | 59.6% |
| | 5 | 1153.35 | 67.3% |
| | 6 | 977.41 | 72.3% |
| | 7 | 849.22 | 75.9% |
| U$_4$-D$_5$ | 1 | 3463.60 | 0 |
| | 2 | 2967.40 | 14.3% |
| | 3 | 1947.06 | 43.8% |
| | 4 | 1476.04 | 57.4% |
| | 5 | 1159.08 | 66.5% |
| | 6 | 1046.54 | 69.8% |
| | 7 | 993.06 | 71.3% |
| U$_4$-D$_4$ | 1 | 3230.49 | 0 |
| | 2 | 2848.91 | 11.8% |
| | 3 | 1823.87 | 43.5% |
| | 4 | 1384.02 | 57.2% |
| | 5 | 1117.19 | 65.4% |
| | 6 | 1024.98 | 68.3% |
| | 7 | 1000.32 | 69.0% |

elastic stage design criterion must be rigorously applied in seismic performance verification. Specifically, the joint rotation angle ($\theta$) should not exceed the yield displacement ($\theta \leq \Delta y$) under frequent earthquake conditions.

The energy dissipation coefficient ($E$) serves as a quantitative metric for evaluating the energy dissipation capacity of the UT-type joint. A comparative analysis of the variation in $E$ across distinct loading stages, as shown in Table 8, reveals that the U$_5$-D$_5$ joint exhibits superior performance, with its average energy dissipation coefficient ($E_{Average}$) consistently exceeding those of the U$_4$-D$_5$ and U$_4$-D$_4$ joints. Notably, $E$ demonstrates progressive enhancement with increasing displacement loading levels for the U$_5$-D$_5$ joint. Conversely, $E$ values for the other two joint types exhibit marginal attenuation during the advanced loading phase. This phenomenon aligns with the degradation trend of the ductility coefficient previously discussed, which is mainly attributed to stiffness degradation induced by buckling instability in the upper connecting plate of the energy-dissipating segment. Based on finite element simulation results, it is recommended that in practical engineering design, the thickness of the upper side plate within the energy-dissipating connection section should satisfy the minimum anti-buckling threshold specified in design codes to ensure the stable energy dissipation performance of the joint under severe seismic events.

## 5.3. Hysteretic behavior of UT-type joints under axial load

Cyclic loading tests conducted at axial force ratios of 0.1, 0.2, and 0.3 demonstrate that the yield moment and yield rotation of UT-type joints are most significantly affected by axial compression. In particular, performance degradation is severe under reverse loading conditions (see Fig 20 for detailed data). Specifically, for the U$_4$-D$_5$ joint, the yield moment and yield rotation decrease by 22.7% and 53.1%, respectively, when the axial force ratio is 0.1. It is worth noting that the buckling resistance of the upper side plate in the U$_4$-D$_5$ joint is the lowest among the three tested joint types. Therefore, in

**Table 8. Average energy dissipation factor of each joint under cyclic loading.**

| Joint number | Load level | *EAverage* (%) | Growth rate (%) |
|---|---|---|---|
| U$_5$-D$_5$ | 1 | 0.04 | 1 |
| | 2 | 0.33 | 7.07 |
| | 3 | 1.01 | 23.38 |
| | 4 | 1.20 | 28.13 |
| | 5 | 1.29 | 30.32 |
| | 6 | 1.35 | 31.74 |
| | 7 | 1.37 | 32.07 |
| U$_4$-D$_5$ | 1 | 0.04 | 1 |
| | 2 | 0.29 | 7.02 |
| | 3 | 0.97 | 25.85 |
| | 4 | 1.20 | 31.97 |
| | 5 | 1.23 | 32.97 |
| | 6 | 1.14 | 30.56 |
| | 7 | 1.07 | 28.46 |
| U$_4$-D$_4$ | 1 | 0.02 | 1 |
| | 2 | 0.27 | 16.16 |
| | 3 | 1.04 | 65.85 |
| | 4 | 1.28 | 81.33 |
| | 5 | 1.39 | 88.83 |
| | 6 | 1.33 | 84.62 |
| | 7 | 1.22 | 77.70 |

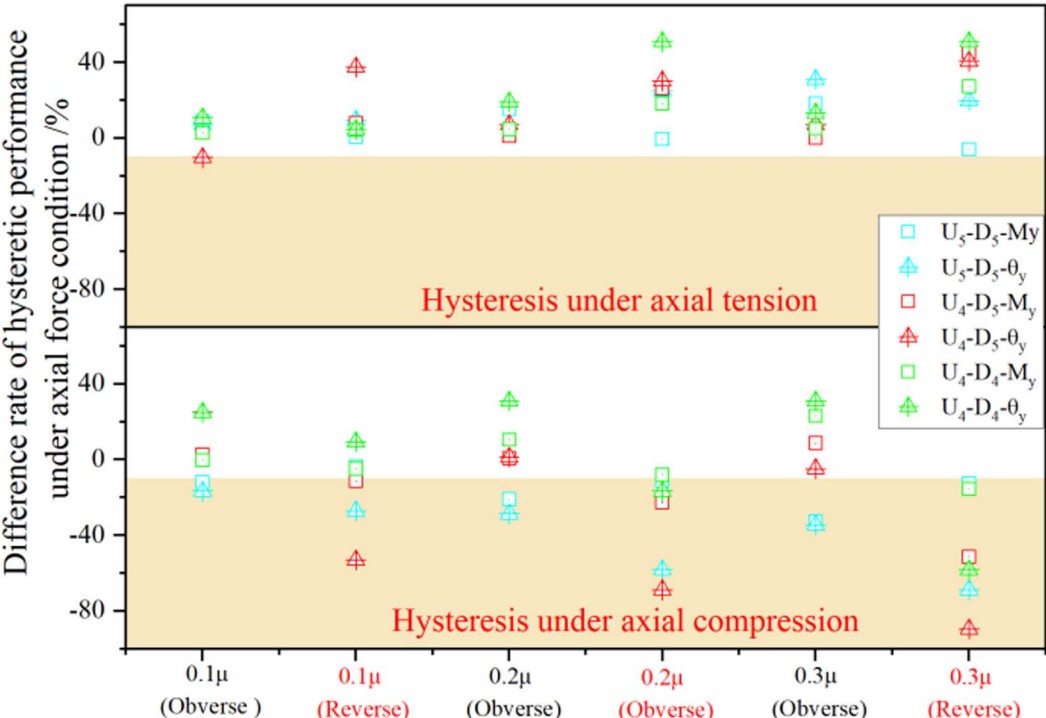

**Fig 20. Difference of characteristic points in hysteretic curves under axial force.**

engineering practice, special attention should be paid to preventing reductions in the buckling resistance of the upper side plates within energy-dissipating connection segments. Additionally, mechanical behaviors of beam-column joints subjected to high axial force ratios require dedicated consideration to ensure structural integrity under seismic demands.

In accordance with the stiffness degradation methodology established for joints without axial loading, the secant stiffness replaces conventional joint stiffness values under cyclic loading. This approach enables calculation of the stiffness degradation curves for each joint under varying axial force ratios. Fig 21 presents the average stiffness variation values across progressive loading stages. Comparative analysis reveals that joints under axial tension and axial compression exhibit fundamentally distinct distributions of stiffness degradation nodes exceeding a 10% average reduction: Under axial tension, significant degradation predominantly occurs during initial loading phases; while under axial compression, pronounced degradation emerges primarily in advanced loading phases. The specific features are as follows:

The initial stiffness decreases significantly under axial tension, mainly due to the early tensile yielding of the upper connecting plate in the energy-dissipating segment. In contrast, stiffness degradation under subsequent loading is more pronounced under axial compression, as a result of more severe buckling of the upper connecting plate in the energy-dissipating segment when the load level increases. Further analysis reveals that under small loading amplitude (stiffness-dominated stage), axial tension causes the upper connecting plate to yield prematurely, thereby reducing the initial stiffness, whereas axial compression can delay yielding or even slightly enhance the initial stiffness. However, during large loading amplitudes (instability-dominated stage), axial compression exacerbates local buckling, leading to a sudden drop in stiffness. This is evident from the data in the table, where the stiffness reduction at each node approaches 70% by the fifth loading stage and exceeds 80% under subsequent loading, indicating a brittle failure trend. Such behavior conflicts with the ductile seismic design principle. Consequently, in engineering applications, the buckling resistance of the upper connecting plate under axial compression demands prioritized attention to preclude non-ductile failure modes.

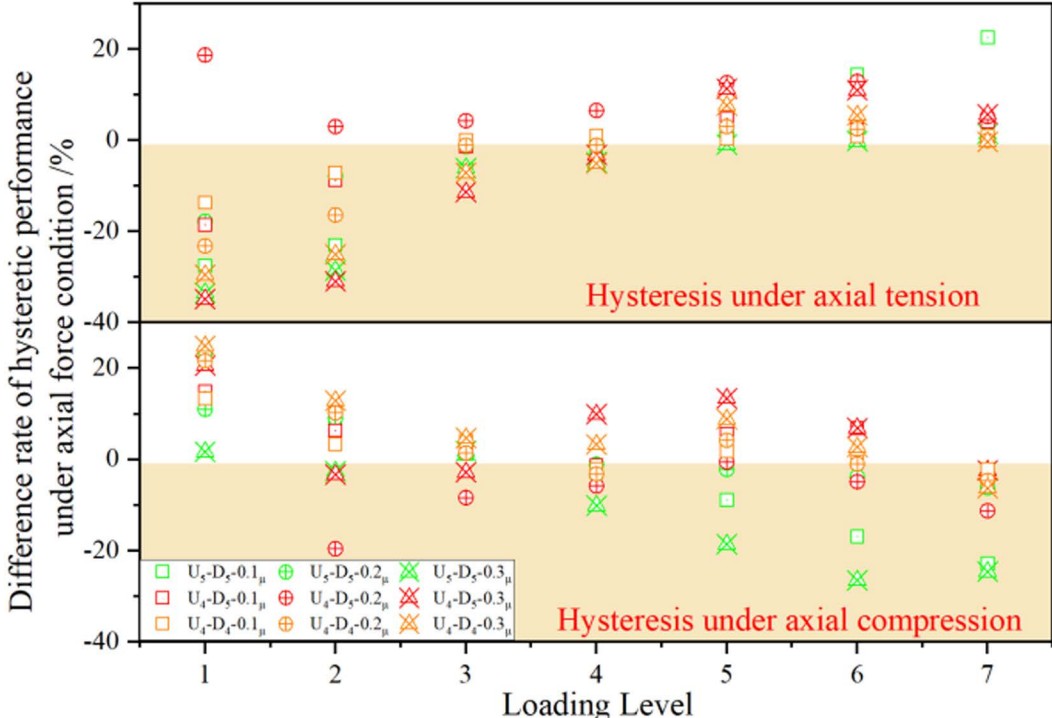

**Fig 21. Average stiffness difference of each joint under axial force.**

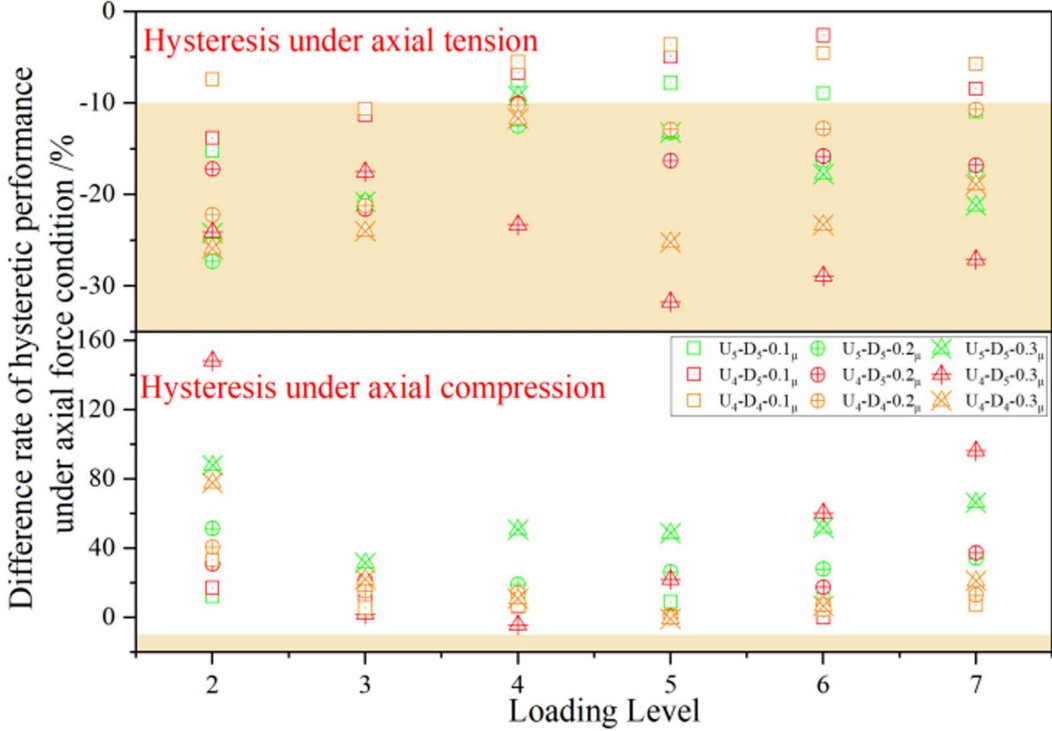

**Fig 22. Difference in average energy dissipation coefficient of each joint under axial force.**

This study investigates the influence of axial force on $E$ performance of UT-type joints by quantifying its variation under different axial force ratios (see Fig 22). The results demonstrate that $E$ decreases significantly under axial tension, with the most pronounced reduction occurring during the stages of large loading amplitude. Conversely, under axial compression, $E$ increases markedly, exceeding 300% during the stages of small load amplitude. This phenomenon is elucidated through hysteresis curve characteristics: axial tension induces a constricted hysteresis loop morphology, whereas axial compression promotes an expanded hysteresis loop morphology. Mechanistic analysis further reveals that under axial compression conditions, the upper side plate of the energy-dissipating segment yields at small loading amplitudes, thus greatly improving energy dissipation efficiency. Under the coupled action of bending moment and axial force, premature slippage of high-strength bolts in the lower region of the joint occurs, resulting in a significant displacement increase after the third stage of loading. This slippage mechanism constitutes the primary cause for the observed reduction in $E$. These findings provide critical insights into the energy dissipation mechanisms of UT-type joints under complex loading scenarios.

It is evident that the axial compression effect on the energy dissipation capacity of UT-type joints appears ostensibly beneficial. Although axial compression significantly enhances the energy dissipation coefficient ($E$) of the UT-type joints during the small-load-amplitude stage (with increases reaching 300%), it concurrently induces pronounced stiffness degradation under large-load-amplitude conditions. This dualistic behavior compromises structural integrity and introduces potential safety hazards. Consequently, $E$ is ill-suited as a primary performance indicator for UT-type joints in engineering applications; instead, priority should be assigned to stiffness degradation characteristics under large-deformation scenarios. To ensure structural safety, implementation of mitigation measures against adverse axial compression effects is imperative, alongside the adoption of a comprehensive performance index that incorporates stiffness degradation metrics for seismic performance evaluation.

## 6. Restoring force model for UT-type joint

In order to facilitate the practical implementation of UT-type joints in engineering structures, a simplified modeling approach for their seismic behavior is imperative. The adoption of solid finite element models entails prohibitively high computational demands, imposing stringent hardware requirements that often exceed the capabilities of conventional engineering design practices. Therefore, establishing a rational simplified restoring force model becomes essential. This model must fulfill three critical criteria: 1) Accurately characterize the key mechanical properties of the joint; 2) Retain sufficient computational accuracy; 3) Significantly improve computational efficiency. Such a simplified framework effectively mitigates the excessive computational resource consumption associated with solid-element modeling, thereby providing a practical analytical foundation for engineering design and assessment.

### 6.1. Restoring force model of joints without axial force effect

For the UT-type joint, the skeleton curve and its characteristic points are extracted from the aforementioned simulation results (as summarized in Table 6). The following assumptions govern the restoring force model: 1) A trilinear model with a descending skeleton curve segment is adopted, corresponding to the specimen's elastic, elastic-plastic, and slip stages; 2) Initial loading follows the skeleton curve, while reverse loading after unloading adheres to the fixed-point pointing rule—directing from the unloading point to the peak displacement point on the skeleton curve. To eliminate the influence of design parameter variations, normalization is applied using the coordinates $(P_y, \Delta_y)$ of the equivalent yield displacement point, where $P_y$ denotes the bearing capacity when the section edge attains the yield stress $f_y$, and $\Delta_y$ represents the displacement at $P_y$. The dimensionless coordinates of the restoring force model are thus defined as $P$ and $\Delta$. Linear regression analysis is performed on the experimental skeleton curve data for each joint, yielding the model's skeleton curve (Fig 23). The fitting equations and slopes for each curve segment are detailed in Table 9.

The unloading stiffness and reloading stiffness of the UT-type joint exhibit progressive degradation with an increasing number of cyclic loading cycles, where the degradation magnitude is governed by the loading phase (elastic or inelastic). Specifically, within the elastic region, the unloading stiffness remains equivalent to the initial elastic stiffness; conversely, in the inelastic region, stiffness progressively diminishes with accumulated plastic deformation. The hysteretic envelope during unloading-reloading cycles is parameterized by stiffness coefficients $K_1 \sim K_4$, as shown in Fig 24. Consistent with the stiffness degradation law derived from hysteresis curves, an exponential function was employed to model the experimental data scatter of unloading and reloading stiffness. The mathematical formulation of this fitting function is detailed in Table 10.

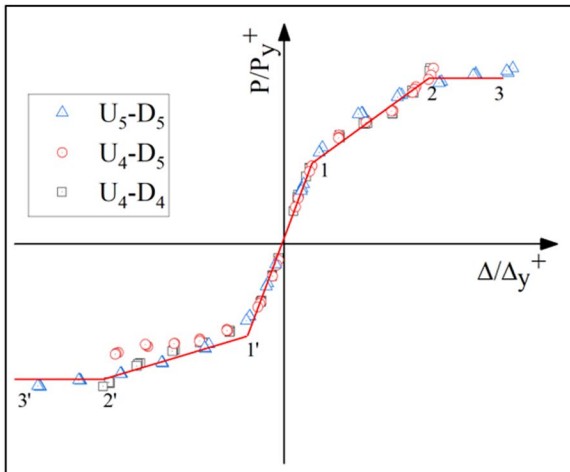

**Fig 23. Fitted skeleton curve.**

**Table 9. Fitting equation of skeleton curve.**

| Loading stage | Function expression | R² |
|---|---|---|
| 1'-1 | $\frac{p}{p_y} = 1.654 \cdot \frac{\Delta}{\Delta_y}$ | 0.99 |
| 1-2 | $\frac{p}{p_y} = 0.398 \cdot \frac{\Delta}{\Delta_y} + 0.401$ | 0.94 |
| 1-2' | $\frac{p}{p_y} 0.205 \cdot \frac{\Delta}{\Delta_y} - 0.393$ | 0.75 |

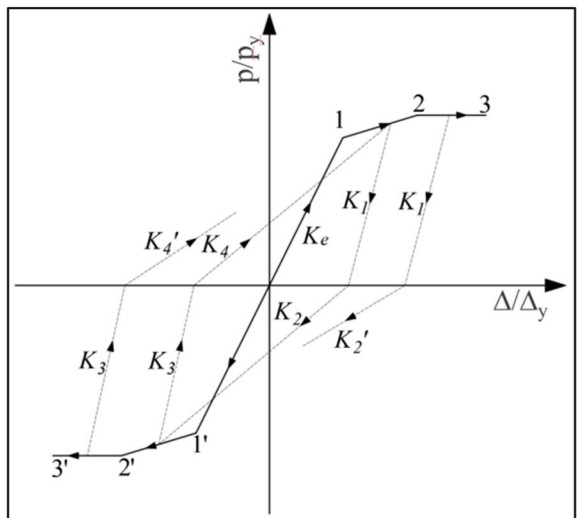

**Fig 24. Restoring force model of UT-type joint.**

**Table 10. Stiffness degradation equation of each stage.**

| No. | Function expression | R² |
|---|---|---|
| $K_1$ | $\frac{K_1}{K_e} = -0.0328 \cdot e^{(\Delta/\Delta_y/0.761)} + 1.072$ | 0.90 |
| $K_2$ | $\frac{K_2}{K_e} = 0.63 \cdot e^{(\Delta/\Delta_y/0.4)} + 0.149$ | 0.83 |
| $K_2'$ | $\frac{K_2'}{K_e} = -0.217 \cdot e^{(\Delta/\Delta_y/6.52)} + 0.439$ | 0.93 |
| $K_3$ | $\frac{K_3}{K_e} = 7.32 \cdot e^{(\Delta/\Delta_y/24.24)} - 6.33$ | 0.91 |
| $K_4$ | $\frac{K_4}{K_e} = 1.499 \cdot e^{(-\Delta/\Delta_y/0.26)} + 0.218$ | 0.92 |
| $K_4'$ | $\frac{K_4'}{K_e} = -0.000163 \cdot e^{(\Delta/\Delta_y/0.34)} + 0.253 \text{ s}$ | 0.93 |

## 6.2. Restoring force model of joints under axial force

The restoring force model for UT-type joints, established through empirical fitting methodologies, exhibits a pronounced dependency on specific axial force conditions. However, axial forces significantly alter the joint's hysteretic characteristics, such as stiffness degradation, yield moment, and yield rotation. Consequently, developing independent models for distinct loading scenarios compromises model versatility and necessitates frequent parameter adjustments in practical applications, thereby reducing computational efficiency. To address these limitations, this paper proposes a simplified restoring force model that explicitly accounts for axial force effects.

First, the yield moment of the UT-type joint is calculated based on the linear stress distribution assumption within the elastic stage, as shown in Figs 25−26 (axial force ratio = 0.3). Analytical results reveal that under axial tension forward loading, the stress at the control point of the joint is mainly governed by the friction of the lower high-strength bolt. Notably, axial tension reduces control point stress (compared to non-axial loading), exerting a positive effect on the yield moment—increasing it by 18.4%. However, the yield moment decreases significantly under three other loading conditions: $U_5$-$D_5$-T-$0.3_\mu$ reverse loading, $U_5$-$D_5$-C-$0.3_\mu$ forward loading, and $U_5$-$D_5$-C-$0.3_\mu$ reverse loading, which decreased the yield moment by 6.4%, 32.4%, and 14.3%, respectively. Critically, this trend remains consistent across diverse axial force ratios, demonstrating its universality. Therefore, modifying the yield moment based on axial force ratios is strongly recommended for adverse working conditions.

Analysis of the yield rotation reveals that the axial tension amplifies the yield rotation ($\theta_{yi}$) of the joint, while the axial compression significantly diminishes this parameter. Adhering to the principle of safety design, the yield rotation is

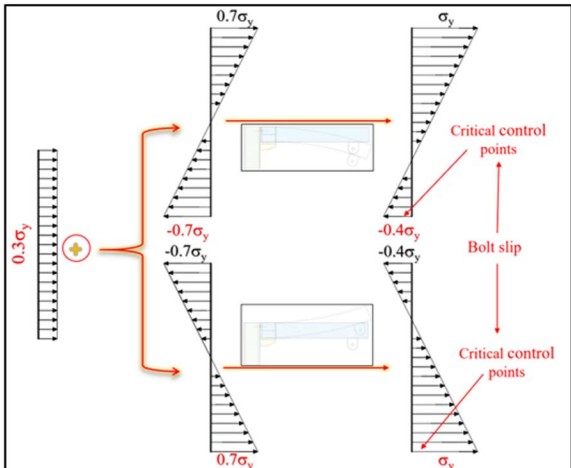

**Fig 25. Stress distribution under axial tension (0.3 μ) in elastic stage.**

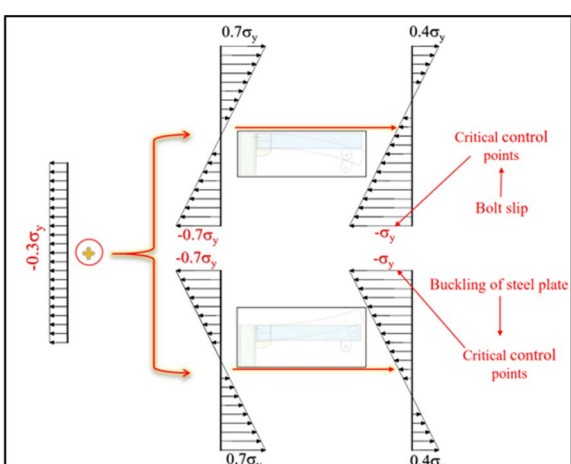

**Fig 26. Stress distribution under axial compression (0.3 μ) in elastic stage.**

quantified by adopting the value under axial tension as equivalent to the no-axial-force condition. Subsequently, the yield angle under axial compression is normalized by calculating $\theta_{yi}/\theta_y$ (where $\theta y$ denotes the mean yield rotation without axial force). This normalization enables the establishment of a quantitative relationship curve between the yield angle variation rate and the axial force ratio (Figs 27–28). Consistent with the most unfavorable design principle in engineering, the lower envelope of the dimensionless yield angle ($\theta_{yi}/\theta_y$) is fitted as the recommended design value for each axial force ratio. This approach provides a conservative reference for the seismic performance evaluation of the joint.

Building upon the methodology for fitting the skeleton curve at the elasto-plastic stage of the joint without axial force (as detailed in Section 6.1), this paper parametrically fits the elastic-plastic stiffness of the skeleton curve under varying axial force conditions. The resulting parameters are summarized in Table 11. For practical engineering applications, linear interpolation is recommended to select the fitting parameters *a* and *b* according to the actual axial force ratio, ensuring computational accuracy and model applicability.

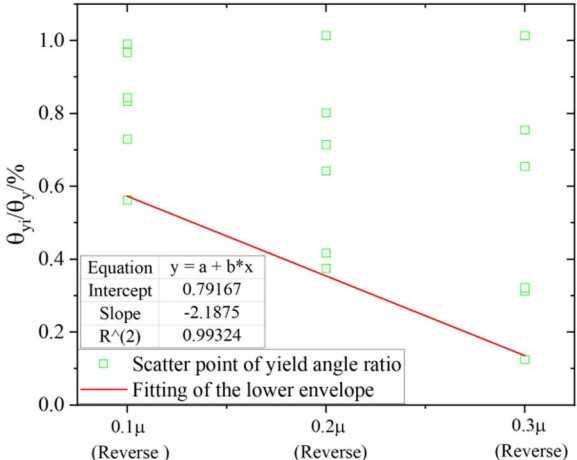

**Fig 27. Ratio of yield rotation under axial force (Obverse loading).**

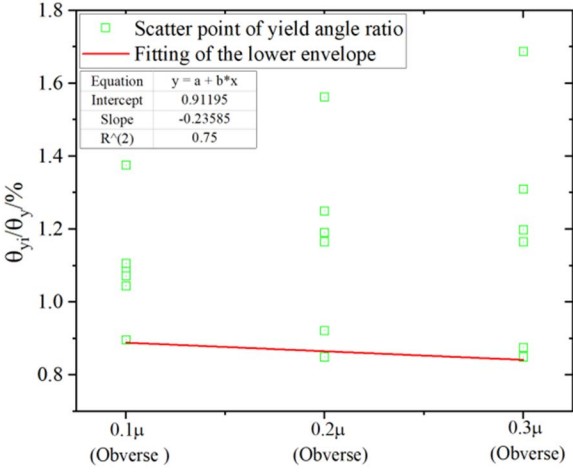

**Fig 28. Ratio of yield rotation under axial force (Reverse loading).**

                                  

**Table 11. Stiffness of joint skeleton curve in elastic-plastic stage under influence of axial force.**

| Joint number | Loading stage | $a$ | $b$ | $R^2$ |
|---|---|---|---|---|
| $U_i$-$D_i$-T-$k_{0.1}$ | 1-2 | 0.669 | 0.324 | 0.83 |
| | 1'-2' | −0.526 | 0.473 | 0.89 |
| $U_i$-$D_i$-C-$k_{0.1}$ | 1-2 | 0.518 | 0.460 | 0.96 |
| | 1'-2' | −0.700 | 0.284 | 0.99 |
| $U_i$-$D_i$-T-$k_{0.2}$ | 1-2 | 0.235 | 1.029 | 0.97 |
| | 1'-2' | −0.362 | 0.620 | 0.95 |
| $U_i$-$D_i$-C-$k_{0.2}$ | 1-2 | 0.389 | 0.564 | 0.89 |
| | 1'-2' | −0.591 | 0.452 | 0.84 |
| $U_i$-$D_i$-T-$k_{0.3}$ | 1-2 | 0.291 | 0.809 | 0.92 |
| | 1'-2' | −0.303 | 0.683 | 0.84 |
| $U_i$-$D_i$-C-$k_{0.3}$ | 1-2 | 0.316 | 0.641 | 0.83 |
| | 1'-2' | −0.944 | 0.053 | 0.85 |

Fitting function model: $p/p_y = a + b \cdot \Delta/\Delta_y$

Parametric analysis of the ultimate moment $M_{ui}$ under varying axial force conditions, when benchmarked against the reference value $M_u$ without axial force (Fig 29), reveals that the reduction in both positive and negative ultimate moments under axial tension is marginally limited to less than 10%. Consequently, the $M_u$ value remains applicable for design scenarios involving axial tension. Conversely, axial compression induces significant degradation of the ultimate bending moment. Therefore, adhering to the most unfavorable design principle, the lower-bound envelope of the axial force ratio versus bending moment ratio ($(M_{ui}-M_u)/M_u$) is fitted to serve as the design basis for the ultimate bending moment under axial compression.

Secondly, the stiffness of the hysteretic envelope during the unloading-reloading process must be quantified (Fig 16). For the unloading stage (segments $K_1$ and $K_3$), the axial force ratio of the beam-column joint in practical engineering typically remains below 0.3. Consequently, axial force ratio exerts negligible influence on the unloading stiffness, justifying

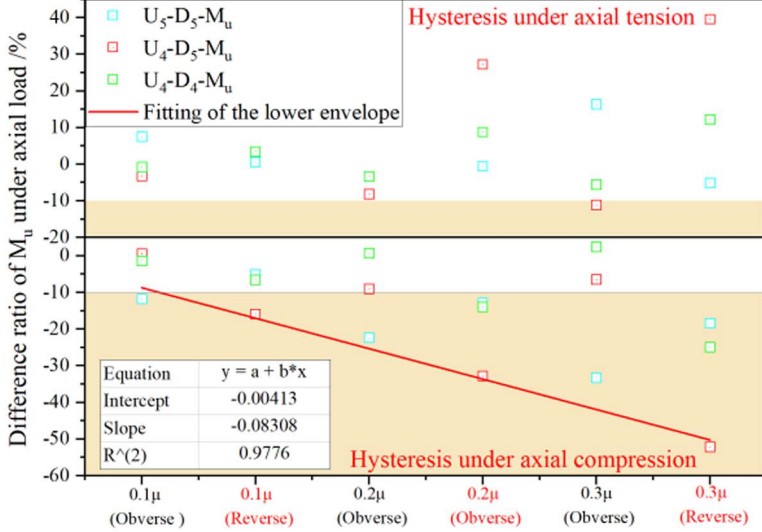

**Fig 29. Bending moment difference ratio ($(Mui-Mu)/Mu$) influenced by axial force.**

the retention of the calculation method for stiffness values without axial force. Regarding the reloading stage, the stiffness parameters ($K_2$, $K_2'$, $K_4$, $K_4'$) are derived from the previously established axial force-dependent skeleton curve, with parameters determined through geometric analytical methods. The derivation process aligns with prior formulations and is not reiterated here.

### 6.3. Verification of joint restoring force model

Comparative analysis between the hysteretic curve derived from the $U_5$-$D_5$ joint simulation and the corresponding restoring force model curve is presented in Fig 30–32. Notably, under no axial force conditions, the stiffness parameters $K_2$, $K_2'$ and $K_4$, $K_4'$ can be directly computed using the aforementioned equations. The resultant curve morphology is represented by a simplified linear connection spanning from the unloading point to the terminus of reverse loading. Comparative assessment reveals that the theoretical model exhibits superior agreement with simulation data in the

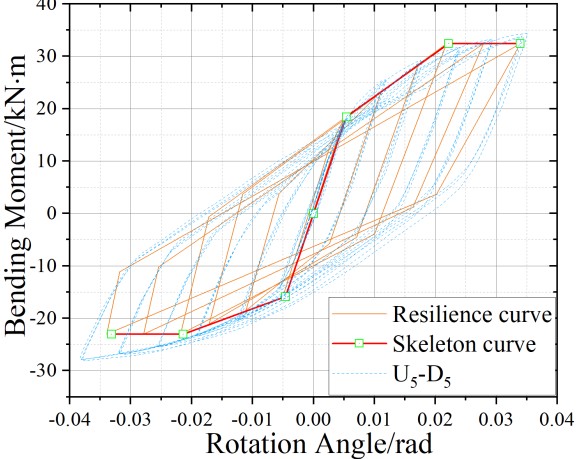

**Fig 30. Restoring force model of joints without axial force effect ($U_5$-$D_5$ as an example).**

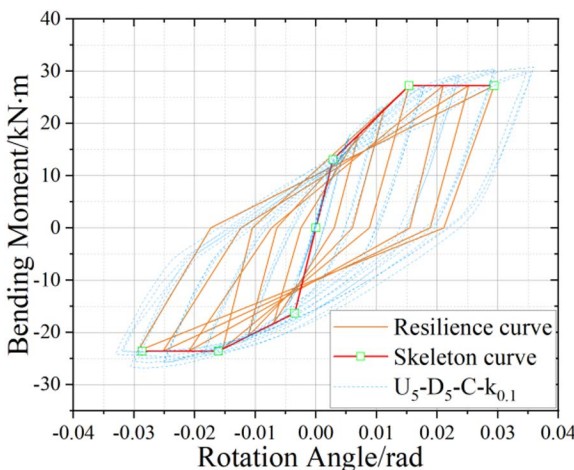

**Fig 31. Restoring force model of joints under axial compression ($U_5$-$D_5$-C-$k_{0.1}$ as an example).**

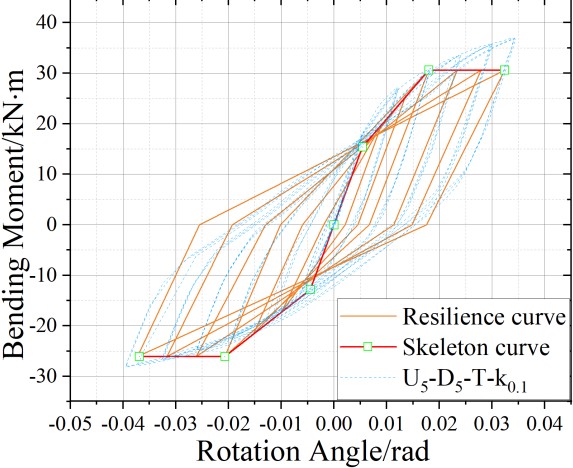

**Fig 32. Restoring force model of joints under axial tensile ($U_5$-$D_5$-T-$k_{0.1}$ as an example).**

absence of axial force, while the theoretical curves for the other two working conditions demonstrate strong correlation with numerical results. To validate the efficacy of the model, the energy dissipation coefficient of the model is quantitatively calculated and benchmarked against simulation outcomes (Table 12). The results indicate that deviations in the energy dissipation coefficient fall within an acceptable threshold, confirming that the theoretical restoring force model accurately characterizes both the load-displacement relationship and energy dissipation behavior of UT-type joints under cyclic loading. This framework provides a robust foundation for predicting the hysteretic response of such joints.

**Table 12. Energy dissipation coefficients of restoring force model.**

| Joint number | Load level | *EAverage* (%) | Error rate (%) |
|---|---|---|---|
| $U_5$-$D_5$ | 2 | 0.21 | 34% |
| | 3 | 1.024 | 1.4% |
| | 4 | 1.26 | 5.0% |
| | 5 | 1.17 | 8.8% |
| | 6 | 0.92 | 31.7% |
| | 7 | 0.84 | 38.7% |
| $U_5$-$D_5$-C-$k_{0.1}$ | 2 | 0.48 | 29.7% |
| | 3 | 1.07 | 1.6% |
| | 4 | 1.05 | 19.2% |
| | 5 | 0.87 | 38.0% |
| | 6 | 0.95 | 35.7% |
| | 7 | 1.06 | 30.3% |
| $U_5$-$D_5$-T-$k_{0.1}$ | 2 | 0.45 | 28% |
| | 3 | 0.89 | 0.8% |
| | 4 | 0.95 | 14.07% |
| | 5 | 0.95 | 19.6% |
| | 6 | 1.00 | 18.3% |
| | 7 | 1.04 | 14.7% |

## 7. Conclusions

In this study, a UT-type prefabricated beam-column joint suitable for rectangular steel tube structures was developed through integrated static loading tests and finite element simulations. The mechanical behavior of the joint under combined axial compression/tension-bending moment was systematically investigated. The key findings are summarized as follows:

1. Structurally, the joint incorporates an adjustable sleeve design to accommodate construction tolerances, thereby resolving alignment discrepancies inherent in on-site assembly. Furthermore, the energy-dissipating connection segment realizes the design objective of "controlled plastification," significantly enhancing the energy dissipation capacity of the joint.

2. Regarding the effect of axial force on the mechanical behavior of the joint, static load tests demonstrate that the axial force significantly influences both the initial stiffness and yield moment of the joint while concurrently reducing its rotation capacity. Specifically, axial compression amplifies the risk of local buckling in the connecting plates within the energy-dissipating segment during reverse loading. Conversely, axial tension induces a characteristic "shrinkage" of the hysteresis loop, thereby diminishing the energy dissipation coefficient ($E$) of the joint. Although axial pressure can substantially enhance the $E$ value, it simultaneously triggers an abrupt stiffness degradation, ultimately elevating the risk of brittle failure mechanisms in the joint.

3. In terms of the energy dissipation mechanism, the energy dissipation capacity of the joint depends critically on the plastic development of the energy-dissipating segment. Furthermore, the segment's section size exerts a significant influence on this capacity. Specifically, the $U_5$-$D_5$ joint (with a thickened upper connecting plate) exhibits superior ductility, as evidenced by an average ductility coefficient of 2.05. Conversely, for the $U_4$-$D_5$ joint (featuring a thinner upper plate), the energy dissipation coefficient ($E$) undergoes a marked decline during the late loading stage due to pronounced buckling. These findings underscore that the connecting plate thickness must satisfy anti-buckling requirements in the design phase to ensure structural resilience.

4. Restoring force model and engineering applicability: A trilinear restoring force model incorporating the influence of axial force is developed, achieving adaptability to varying axial force ratios through parametric adjustments to its skeleton curve and stiffness degradation law. Verification analyses based on the energy dissipation coefficient demonstrate that the theoretical model exhibits close agreement with simulation results, with the error remaining within an acceptable engineering tolerance range.

## Author contributions

**Funding acquisition:** Menghan Sun.

**Methodology:** Luyao He.

**Writing – review & editing:** zailin Yang.

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
