## [Decision Letter · Decision Letter 0]

PONE-D-25-27560Characteristics of a new UT-Type Prefabricated Joint for Rectangular Steel Tubes under Combined Axial-Moment LoadingPLOS ONE

Dear Dr. Yang,

Thank you for submitting your manuscript to PLOS ONE. After careful consideration, we feel that it has merit but does not fully meet PLOS ONE’s publication criteria as it currently stands. Therefore, we invite you to submit a revised version of the manuscript that addresses the points raised during the review process.

We look forward to receiving your revised manuscript.

Kind regards,

Dajiang Geng

Academic Editor

PLOS ONE

“This work was supported by the National Natural Science Foundation of China (Grant No. 52279128) and Heilongjiang Provincial Natural Science Foundation (Grant No.YQ2022E013).”

4. Thank you for stating the following in the Funding Section of your manuscript:

“This work was supported by the National Natural Science Foundation of China (Grant No. 52279128) and Heilongjiang Provincial Natural Science Foundation (Grant No.YQ2022E013).”

“This work was supported by the National Natural Science Foundation of China (Grant No. 52279128) and Heilongjiang Provincial Natural Science Foundation (Grant No.YQ2022E013).”

5. We note that your Data Availability Statement is currently as follows: [All relevant data are within the manuscript and its Supporting Information files.]

Reviewers' comments:

Reviewer's Responses to Questions

**Comments to the Author**

1. Is the manuscript technically sound, and do the data support the conclusions?

Reviewer #1: Yes

Reviewer #2: Yes

2. Has the statistical analysis been performed appropriately and rigorously? 

Reviewer #1: Yes

Reviewer #2: Yes

3. Have the authors made all data underlying the findings in their manuscript fully available?

Reviewer #1: Yes

Reviewer #2: Yes

4. Is the manuscript presented in an intelligible fashion and written in standard English?

Reviewer #1: Yes

Reviewer #2: Yes

5. Review Comments to the Author

Reviewer #1: A comprehensive study on the static and hysteretic behavior of a new type of Prefabricated beam-column joint (UT-type) is presented in the manuscript entitled “Characteristics of a new UT-Type Prefabricated Joint for Rectangular Steel Tubes under Combined Axial-Moment Loading”, in order to promote the practical application of the joint, a simplified trilinear restoring force model is proposed. The quality of the paper is good, but some parts need to be improved:

1. Joint Construction and Assembly Method: The technological explanation in Section 2 regarding the assembly of the UT-type joint components, particularly the bolt tube and the threaded sleeve, requires clearer and more detailed elaboration.

2. Experimental Methodology: he justification for omitting the 0.2μ and 0.6μ failure modes in Section 3.1 remains incomplete. To address this gap, the text should clarify that "Results for μ=0.2/0.6 are omitted here due to similar failure patterns but included in Table 3 for quantitative comparison." This addition would provide readers with a clear rationale for the selective data presentation while ensuring all critical information remains accessible in the supplementary data tables. 3 for quantitative comparison."

3.Abstract Structure: The novelty and key quantitative findings of the study are not prominently highlighted. To improve clarity, the abstract should explicitly state: *"We propose a UT-type prefabricated joint featuring adjustable sleeves (tolerance: ±5 mm) and energy-dissipating segments. Tests reveal axial tension increases initial stiffness by 42.6% (μ=0.4), while compression reduces yield moment by 24.5%. The validated trilinear model predicts hysteresis with <10% error."* This revision ensures the innovation and critical results are immediately apparent to readers.

4. Figures & Tables (Moderate Revisions for Readability & Rigor):Some tables in the article are missing units, such as table 7 and Table 8; Some figures (e.g., Figs. 6–7) would benefit from clearer labeling and more informative captions to help readers better understand the observed failure modes.

5.Improve readability and adherence to academic writing standards, please revise passive voice constructions to active voice (e.g., change "It was observed that..." to "We observed..." in lines 78 and 112) for greater directness and clarity, as recommended by most style guides. Additionally, correct unit spacing format (e.g., "3.7mm" should be written as "3.7 mm" in Table 2) to follow standard SI unit conventions and maintain consistency throughout the manuscript. These modifications will enhance both the precision and professionalism of the text.

6. Nomenclature Consistency: The current manuscript alternates between the terms "energy-dissipating" (line 45) and "energy-consuming" (line 12), which may cause confusion. For standardization and compliance with ASTM terminology, all instances should be unified under the preferred term "energy-dissipating". This correction ensures technical precision and maintains consistency with established industry standards throughout the paper.

7. The following research works regarding the prefabricated beam-column joints and related structures can be reviewed and are suggested for possible reference:

(1) Modeling of force- displacement behavior of post- tensioned self-centering concrete connections. Engineering Structures, 2019, 198: 109538.

(2) Probabilistic analysis on time-dependent mechanical behavior of posttensioned self-centering concrete connections. Engineering Structures, 2020, 218, 110856.

(3) Influence of connection properties on seismic performance of post-tensioned self-centering concrete frames, Journal of Building Engineering, 2022, 46, 103761.

Reviewer #2: The manuscript presents an investigation into the UT-type prefabricated beam-column joint, combining experimental testing (static and cyclic loading) with finite element simulations to analyze its mechanical behavior under axial force-bending moment coupling. The inclusion of both numerical and experimental work is a strong foundation, and the experimental program—particularly the cyclic loading tests—provides valuable insights into energy dissipation and failure modes, which are critical for engineering applications. The proposed trilinear restoring force model adds practical value by offering a design-oriented tool. However, several fundamental issues regarding clarity of objectives, novelty of the joint design, and technical rigor require major revisions to strengthen the study's contribution to the field.

A A critical gap in the manuscript is the lack of clarity on whether the UT-joint represents a new design innovation or an adaptation of existing prefabricated connections.

B The research objectives are not explicitly articulated, leading to a disjointed narrative between the problem statement, methodology, and findings. While the study explores mechanical behavior and proposes a restoring force model, the overarching goals—such as improving seismic performance of prefabricated structures, addressing constructability challenges, or enhancing design codes—are not clearly defined.

C The writing of the manuscript requires substantial improvement. The language is often unclear, and the sentences are poorly structured, making it difficult for readers to follow the arguments. The authors should pay more attention to grammar, punctuation, and sentence construction.

D Format and grammar Issues requiring correction:

1. The abbreviation UT-Type is never defined.

2 The reference style should be consistent. References mix author-year and numeric citation formats. If the reference number is used, there is no need to include year. And usually, the last name is used when referring.

3 Some paragraphs are too short and only have one sentence (I.e. Line 66, 73). Please combine to improve the flow.

4 No unit in Figure 4

5 Line showed Eq. (3), where is Eq. (1) and (2)

6 The format of tables should be consistent (i.e. Line space, space between number and unit)

7 Ensure all technical symbols are italicized throughout the manuscript.

8 What is the arrows in Table 3 represent for?

9 Line 96 is not a new sentence; there is no need of the indentation and uppercase of W.

6. PLOS authors have the option to publish the peer review history of their article (what does this mean? ). If published, this will include your full peer review and any attached files.

**Do you want your identity to be public for this peer review?** For information about this choice, including consent withdrawal, please see our Privacy Policy .

Reviewer #1: No

Reviewer #2: No

---

## [Author Response · Author response to Decision Letter 1]

25 Jun 2025

Response to Reviewer Comments

Reviewer #1

(1).** Reviewer’s Comment:**

Joint Construction and Assembly Method: The technological explanation in Section 2 regarding the assembly of the UT-type joint components, particularly the bolt tube and the threaded sleeve, requires clearer and more detailed elaboration.

**Response:**

We appreciate the reviewer's careful reading and constructive suggestion. In accordance with the review comments, we have thoroughly revised Section 2.1 ( lines 69-84 ), focusing on enhancing the technical details regarding the joint installation methods, including the prefabrication steps, installation steps, and precautions. The modified content has been highlighted in the text; please verify whether it meets your requirements.

(2).** Reviewer’s Comment:**

Experimental Methodology: he justification for omitting the 0.2μ and 0.6μ failure modes in Section 3.1 remains incomplete. To address this gap, the text should clarify that "Results for μ=0.2/0.6 are omitted here due to similar failure patterns but included in Table 3 for quantitative comparison." This addition would provide readers with a clear rationale for the selective data presentation while ensuring all critical information remains accessible in the supplementary data tables. 3 for quantitative comparison."

**Response:**

Thank you for your review comments. We have revised Section 3.1 to explicitly state: "Results for μ=0.2 and 0.6 exhibit similar failure patterns to μ=0.4 and are omitted here for conciseness. Their quantitative data are provided in Table 3 and Figure 5 for direct comparison."

(3).** Reviewer’s Comment:**

Abstract Structure: The novelty and key quantitative findings of the study are not prominently highlighted. To improve clarity, the abstract should explicitly state: *"We propose a UT-type prefabricated joint featuring adjustable sleeves (tolerance: ±5 mm) and energy-dissipating segments. Tests reveal axial tension increases initial stiffness by 42.6% (μ=0.4), while compression reduces yield moment by 24.5%. The validated trilinear model predicts hysteresis with <10% error."* This revision ensures the innovation and critical results are immediately apparent to readers.

**Response:**

We appreciate the reviewer's careful reading and constructive suggestion. Based on the review, we have revised most of the abstract, "We propose a UT-type prefabricated joint featuring adjustable sleeves (tolerance: ±5 mm) and energy-dissipating segments. Tests reveal axial tension increases initial stiffness by 42.6% (μ=0.4), while compression reduces yield moment by 24.5%. The validated trilinear model predicts hysteresis with <10% error."* has been included in the new Abstract.

(4).** Reviewer’s Comment:**

Figures & Tables (Moderate Revisions for Readability & Rigor):Some tables in the article are missing units, such as table 7 and Table 8; Some figures (e.g., Figs. 6–7) would benefit from clearer labeling and more informative captions to help readers better understand the observed failure modes.

**Response:**

We appreciate this valuable comment. In accordance with the review comments, we have supplemented the corresponding units in Tables 7 and 8, and systematically verified all other tables and illustrations throughout the text to ensure that all measurement units are fully indicated. We have revised Figs. 6-7 with improved labeling (e.g., highlighted failure modes ) and enhanced captions to clarify key observations. The modifications enhance readability and ensure better interpretation of the results.

(5).** Reviewer’s Comment:**

Improve readability and adherence to academic writing standards, please revise passive voice constructions to active voice (e.g., change "It was observed that..." to "We observed..." in lines 78 and 112) for greater directness and clarity, as recommended by most style guides. Additionally, correct unit spacing format (e.g., "3.7mm" should be written as "3.7 mm" in Table 2) to follow standard SI unit conventions and maintain consistency throughout the manuscript. These modifications will enhance both the precision and professionalism of the text.

**Response:**

We thank the reviewer for this suggestion. The entire manuscript has been professionally proofread and polished (by professional writer for clarity, consistency, and grammatical accuracy). All minor language issues, including grammar, syntax, and formatting inconsistencies, have been corrected to improve readability.

(6).** Reviewer’s Comment:**

Nomenclature Consistency: The current manuscript alternates between the terms "energy-dissipating" (line 45) and "energy-consuming" (line 12), which may cause confusion. For standardization and compliance with ASTM terminology, all instances should be unified under the preferred term "energy-dissipating". This correction ensures technical precision and maintains consistency with established industry standards throughout the paper.

**Response:**

Thank you for your review comments. The entire text has been revised from energy-consuming to energy-dissipating in accordance with the review comments.

(7).** Reviewer’s Comment:**

The following research works regarding the prefabricated beam-column joints and related structures can be reviewed and are suggested for possible reference:

(1) Modeling of force- displacement behavior of post- tensioned self-centering concrete connections. Engineering Structures, 2019, 198: 109538.

(2) Probabilistic analysis on time-dependent mechanical behavior of posttensioned self-centering concrete connections. Engineering Structures, 2020, 218, 110856.

(3) Influence of connection properties on seismic performance of post-tensioned self-centering concrete frames, Journal of Building Engineering, 2022, 46, 103761.

**Response:**

We sincerely appreciate the reviewer’s thoughtful suggestion to contextualize our work within recent advancements in stress-driven and multi-material joint design. As requested, we have carefully reviewed and cited the recommended references to better align our study with state-of-the-art methodologies. These citations now appear in our References section (see citations [16]–[18]).

Reviewer #2

(1).** Reviewer’s Comment:**

A critical gap in the manuscript is the lack of clarity on whether the UT-joint represents a new design innovation or an adaptation of existing prefabricated connections.

**Response:**

We sincerely appreciate the reviewer's insightful observation regarding the novelty of the UT-joint. To clarify, the UT-joint is indeed an original design innovation rather than an adaptation of existing prefabricated connections.

As explicitly stated in the first paragraph of Section 2.1:

The UT-joint employs an all-bolted connection mechanism, fundamentally differing from conventional welded joints;

It’s geometrical configuration and construction details (Illustrated in Fig. 1) are distinctly novel compared to traditional joint systems.

(2).** Reviewer’s Comment:**

The research objectives are not explicitly articulated, leading to a disjointed narrative between the problem statement, methodology, and findings. While the study explores mechanical behavior and proposes a restoring force model, the overarching goals—such as improving seismic performance of prefabricated structures, addressing constructability challenges, or enhancing design codes—are not clearly defined.

**Response:**

Thank you for highlighting the need for clearer research objectives. In the revised manuscript, we have restructured the introduction to provide a more cohesive narrative. The key objectives and motivations of this study are now explicitly outlined as follows:

1. Development a joint (UT-joint):

We first propose a new bolted connection joint (UT-joint), which differs significantly from traditional welded joints in both its connection method (fully bolted) and structural configuration.

2. Mechanical behavior analysis under axial force:

Given the engineering demand for such joints, we investigate the structural performance of the UT-joint under axial loading.

3. Simplified trilinear model for engineering applications:

Finally, to facilitate the use of this joint in structural analysis software, we establish a simplified restoring force model.

4. Ultimate goal:

The overall research aims to promote the practical application of the UT-joint in real-world engineering projects by providing theoretical and numerical support for its design and analysis.

(3).** Reviewer’s Comment:**

The writing of the manuscript requires substantial improvement. The language is often unclear, and the sentences are poorly structured, making it difficult for readers to follow the arguments. The authors should pay more attention to grammar, punctuation, and sentence construction.

**Response:**

We thank the reviewer for this suggestion. The entire manuscript has been professionally proofread and polished (by professional writer for clarity, consistency, and grammatical accuracy). All minor language issues, including grammar, syntax, and formatting inconsistencies, have been corrected to improve readability.

(4).** Reviewer’s Comment:**

Format and grammar Issues requiring correction:

1. The abbreviation UT-Type is never defined.

2 The reference style should be consistent. References mix author-year and numeric citation formats. If the reference number is used, there is no need to include year. And usually, the last name is used when referring.

3 Some paragraphs are too short and only have one sentence (I.e. Line 66, 73). Please combine to improve the flow.

4 No unit in Figure 4

5 Line showed Eq. (3), where is Eq. (1) and (2)

6 The format of tables should be consistent (i.e. Line space, space between number and unit)

7 Ensure all technical symbols are italicized throughout the manuscript.

8 What is the arrows in Table 3 represent for?

9 Line 96 is not a new sentence; there is no need of the indentation and uppercase of W.

**Response:**

We sincerely appreciate the reviewer’s meticulous comments, which have helped enhance the manuscript’s clarity and consistency. All noted issues have been addressed as follows:

We have defined UT-Type in Figure 1, as detailed in Section 2.1. (etymologically deriving its “UT” designation from the cross-sectional profile resembling said letters; see Figure 1).

We have revised the reference style of the entire text in accordance with the comments, ensuring its uniformity.

We confirm that all single-sentence paragraphs (e.g., original Lines 66 and 73) have been thoroughly revised. These standalone sentences have now been logically integrated into adjacent paragraphs to improve textual cohesion and readability. The manuscript’s narrative flow has been carefully checked to ensure no fragmented paragraphs remain.

The corresponding units of length have been added in Figure 4.

We sincerely apologize for the oversight in the equation numbering sequence. The manuscript has now been thoroughly checked and corrected to ensure proper numbering continuity. All subsequent cross-references to equations have also been verified for consistency.

We thank the reviewer for highlighting this important formatting issue. All tables have now been carefully standardized to ensure: Consistent line spacing throughout, uniform spacing between numerical values and their units. The formatting has been checked to guarantee full consistency across the manuscript.

We have carefully checked and corrected all technical symbols (e.g., Rm, ReL, ReH) to ensure consistent italicization throughout the manuscript, as per standard conventions.

We have explained the meaning of the arrows in Table 3 (\ (↓^\ast)\ represents the ratio of the increase in theoretical data compared to experimental data. {(\downarrow}^\ast) represents the ratio of the decrease in theoretical data compared to experimental data.).

We have checked and corrected this issue (removed the unnecessary indentation and lowercase "w" in Line 96). Additionally, we have reviewed the entire manuscript to ensure consistent formatting in similar cases.

---

## [Editor Report · Decision Letter 1]

Characteristics of a new UT-Type Prefabricated Joint for Rectangular Steel Tubes under Combined Axial-Moment Loading

PONE-D-25-27560R1

Dear Dr. zailin Yang,

We’re pleased to inform you that your manuscript has been judged scientifically suitable for publication and will be formally accepted for publication once it meets all outstanding technical requirements.

Kind regards,

Dajiang Geng

Academic Editor

PLOS ONE
---

## [Editor Report · Acceptance letter]

PONE-D-25-27560R1

PLOS ONE

Dear Dr. Yang,

I'm pleased to inform you that your manuscript has been deemed suitable for publication in PLOS ONE. Congratulations! Your manuscript is now being handed over to our production team.

Kind regards,

on behalf of

Dr. Dajiang Geng

Academic Editor

PLOS ONE